# Adsorption to soils and biochemical characterization of purified phytases

María Marta Caffaro [1,2]; Karina Beatriz Balestrasse [1,3]; Gerardo Rubio [1,2]

[1] INBA, CONICET UBA, Buenos Aires, C1417DSE, Argentina
[2] Soil Fertility and Fertilizers, School of Agriculture University of Buenos Aires, Buenos Aires, C1417DSE, Argentina
[3] Biochemistry, School of Agriculture University of Buenos Aires, Buenos Aires, C1417DSE, Argentina

*Correspondence to*: Gerardo Rubio (rubio@agro.uba.ar)

**Abstract.** Four purified phytases isolated from *Aspergillus niger* and *Escherichia coli* were characterized biochemically and in terms of their adsorption to soils belonging to the Mollisol order. Three different organic P substrates were used to measure enzyme activity in a wide range of pH (2.3 to 9) and temperatures (-10º to 70ºC): p-nitrophenyl-phosphate, glyceraldehyde-3-phosphate and phytic acid. Phytases from *A. niger* showed a higher capacity to release P (36 to 50% of P contained in the substrates, 44 to 62 µg P), than phytases from *E. coli* (24 to 36%, 20 to 44 µg P). The amount of P released from organic P substrates by *A. niger* phytases followed the following range: p-nitrophenyl-phosphate > glyceraldehyde-3-phosphate > phytic acid whereas in *E. coli* phytases the order was p-nitrophenyl-phosphate/glyceraldehyde-3-phosphate > phytic acid. All phytases were active throughout the pH and temperature ranges for optimum crop production. The proportion of phytases found in the solid phase of the soil 60 minutes after addition was lower than that found in the liquid phase (23-34% vs. 66-77%). Obtained results are promising in terms of the use of phytases as a complement to P fertilization in agricultural settings and encourages further studies under field conditions.

## 1 Introduction

Phosphorus (P) is the second nutrient that limits agricultural crop productivity worldwide. Most strategies for enhancing P nutrition of agricultural crops aim to maintain soils at the appropriate P critical level so that yields are not constrained by this nutrient and environmental pollution risks are avoided or minimized (Simpson et al. 2011). The most widely used practice to overcome soil P deficiencies is the application of inorganic P fertilizers produced from phosphate rock (PR). However, world PR reserves that can economically be extracted are estimated to be depleted in the next 50-100 years (Cordell et al., 2009). Several strategies have been suggested to increase P utilization efficiency and reduce PR-derived fertilizers consumption (Cornish, 2009; Richardson et al., 2009; Veneklaas et al., 2012; Fernandez and Rubio, 2015). Richardson et al. (2011) summarizes these strategies in three groups: i) root-foraging strategies that reduce the critical P requirements for plant growth; ii) P-mining strategies that enhance the P availability from sparingly-available sources in soil, and iii) physiological strategies that lead to higher yields per unit of P uptake.

Soil P is comprised of inorganic and organic forms. Phosphates present in the soil solution are the main source of P for higher plants. Due to the strong interaction with the soil matrix, concentration of phosphates in the soil solution is very low (<10µM) (Dalal, 1977). In general, soil organic P content varies in a wide range (between 30-70% of total soil P; Condron et al., 2005; Nash et al., 2014; Cabello et al., 2016; and even on a wider range, Harrison 1987). The predominant soil organic P fractions are usually phytates (Harvey et al., 2009; Steffens et al., 2010), followed by nucleic acids, phospholipids and sugar-phosphates (Tiessen, 2008). Phytates and other organic P forms cannot be directly utilized by plants and need to be mineralized before being ready for plant uptake. The discovery of phytate-degrading compounds changed the conventional perception that phytate was a recalcitrant molecule in the environment (Condron et al., 2005; Harvey et al., 2009).

Phytases are enzymes released by bacteria, fungi, plants and animals (Jorquera et al., 2008) and are able to catalyze the release of P from phytates. Although phytases are distributed throughout the soils, the higher concentrations are found in the rhizosphere (Li et al., 2008). The first phytase was discovered in the early 20th century (Hill and Richardson, 2006), but their precise identification was not made until the mid-60s (Tabatabai and Bremmer, 1969). Phytases are proteins widely distributed in soil microorganisms (Mullaney and Ullah, 2007). The high capacity of A. niger and E. coli to secrete phytases has promoted their use as a source of these enzymes in commercial production by the industry (Misset, 2003). A. niger phytases are mainly extrinsic (Azeem et al., 2015), and are classified as 3-phytases, because they primarily dephosphorylate the phosphate group located at 3-position. E. coli phytases are mainly membrane-associated proteins and were classified as 6-phytase (Azeem et al., 2015). The classification as 3- or 6-phytases is related to which phosphate group is attacked first and would be determined by conformational differences in the β-domain of each phytase (Konietzny and Greiner, 2002).

Besides being a key fraction of soil organic P, phytates are present in other nature components. For example, 60-90% of P in cereal and oil grains is present under phytic acid forms (63% in soybean, 77% in wheat, 83% in maize; Lott et al., 2000). One of the most common uses of these grains is for livestock feed (Misset, 2003). However, the microbial population of the digestive tract of monogastric animals (e.g. poultry) is unable to utilize phytate as a P source. The benefit of adding phytases to poultry diet to enhance phytic acid P utilization was demonstrated some time ago and nowadays is a widespread practice in poultry nutrition management (El-Sherbiny et al., 2010). It was demonstrated that using phytases from different microorganisms (i.e. Aspergillus spp and *Escherichia coli*) for this practice may also entail environmental benefits by reducing the P content of poultry manure.

Extensive use of phytases in livestock and aquaculture production contrasts with the practically null use in agriculture. There are very few reports in which phytases were studied to enhance soil P availability (e.g. Findenegg and Nelemans, 1993; Gaind and Nain, 2015, Liu et al., 2018). Adding phytases to poor P soils increased biomass accumulation of maize by around 32% (Findenegg and Nelemans, 1993). Undoubtedly, phytase research appears to be a promising path to increase soil P use efficiency (Büneman, 2008; Gaind and Nain, 2015; Menezes-Blackburn et al., 2016; Liu et al., 2018). Some reports indicate that the adsorption of phytases to the soil matrix may reduce their affinity for substrates containing P (George et al., 2005; Giaveno et al., 2010; Yang and Chen, 2017). When pH increases, clay charge changes, decreasing the phytase affinity (Ruyter-Hooley et al., 2015).

In this work we evaluated the performance of four commercially available phytases, two extracted from *Aspergillus niger* and two from *Escherichia coli* as candidates to be used as a biological fertilizer to release inorganic P from organic P sources. Our working hypotheses were: i) phytases have the ability to release P from different organic P sources, with preference for phytic acid; ii) the retention of phytases in the soil solid phase is associated to the soil clay content; iii) the two evaluated phytases differ in the pH and temperature levels to reach their optimum activity.

## 2 Materials and methods

### 2.1 Enzyme preparation

Four phytases were used in our experiments; two isolated from A. niger of two different batches of Habio (Sichuan Habio Bioengineering Co., Ltd) here named *A. niger* 1 and 2 and two isolated from *E. coli* (TS Smizyme phytase, Quimtia EDF and Ronozyme, DSM) named *E. coli* 1 and 2, respectively. These enzymes are in powder format at a concentration of 5000 U g-1. Two hundred mg of each phytase were suspended in a solution composed by 20 ml of 360 mM $CaCl_2$, 1 mM buffer pH 5.5 sodium acetate, and 100 mg g-1 Tween 20. The solution was mixed 30 min at

4 °C and subsequently centrifuged at 6900 g for 30 min at the same temperature. Final concentration of enzymes in the solution was 10 mg enzyme $ml^{-1}$.

**2.2 Phytase adsorption on soils**

Soil samples (0-20 cm) were taken from seven representative soils of the Pampean Region, the most productive area of Argentina (Table 1). All soils belong to the Mollisol order (Rubio et al. 2019). One gram of each soil and 20 ml of phytase solution (17.6 nKat $g^{-1}$ of soil, specific activity 8.3 nKat $mg^{-1}$ protein) was placed in 50 ml screw-capped polyethylene tubes at room temperature (22 ºC). After shaking the tubes on a flat bed shaker (75 oscillations $min^{-1}$) sub-samples of soil slurry (500 ml) were taken for phytase activity measurements at 5, 10, 15, 30 and 60 min. To obtain a representative sample of the suspension, aliquots of soil slurry were taken using a pipette tip after vigorously mixing the soil suspension. An aliquot (150 ml) of the soil slurry was used to measure the enzyme activity (here called soil suspension). The remainder portion of the sample was centrifuged at 15,000 g for 5 min and the supernatant was taken for measuring the phytase activity (called soil solution).

Phytase activities in aliquots of soil solutions and suspensions were measured at a 1:1 sample to buffer ratio. Assays were performed against phytic acid substrate for 60 min at 37°C at a final concentration of 2 mM and buffered to pH 5.5 with 15 mM MES (George et al., 2005). Reactions were stopped with an equal volume of 10% TCA. Samples were centrifuged at 3800g for 5 min prior to determination of P concentration in the supernatant using Murphy-Rilley method (Murphy and Riley, 1962). Phytase activity retained in the solid phase was determined by calculating the difference between the phytase activity of the soil suspension and activity of the soil solution. To determine which soil characteristics (Table 1) affected phytase distribution between soil solid and liquid phases, a linear regression and correlation analysis between $y_{max}$ (maximum distribution of the enzyme in the soil solid phase) and k (rate at which distribution peaks) with soil characteristics were performed.

**2.3 Biochemical characterization, pH and temperature optimum levels**

Biochemical characterization of the phytases included: total protein by Lowry method (Lowry et al., 1951), enzymatic activity as a function of pH and temperature, kinetic parameters $V_{max}$ and $K_m$ and adsorption to seven selected soils.

Phytase activity was measured with three substrates containing 10 mM P: 2 mM phytic acid, 10 mM p-nitrophenyl-phosphate and 10 mM glyceraldehyde-3-phosphate. In this experiment incubation temperature was 25 °C according to Hayes et al. (1999).

To evaluate the performance of the enzymes along a pH range (2.3-9.0), 200 µl of each enzyme solution was diluted with 400 µl of 50 mM glycine-HCl buffer (pH 2.3-4.4), 50 mM Na-acetate (pH 3.6-5.8), 50mM MES-KOH (pH 5.2-7.3) and 50 mM Tris-HCl (pH 6.1-9.0), as a reaction buffer. To evaluate the performance of the enzymes along a temperature range (-10-70ºC), 200 µl of each enzyme solution was diluted with 400 µl MES (pH 5.5) buffer. For both pH and temperature studies, incubation time was 1 h and the reaction was terminated by the addition of 10% trichloroacetic acid (TCA). In the temperature studies, the buffer containing the substrates is heated until the desired temperature is reached. At this point the enzyme is added and the incubation time starts Measurements were performed in triplicate. The activities were tested against three blanks: blank 1: reaction buffer without enzyme or substrate; blank 2: reaction buffer with enzyme without substrate; and blank 3: reaction buffer without enzyme with substrate. When the substrates were phytic acid and glyceraldehyde-3-phosphate, phytase activity was measured by the Murphy-Riley method (Murphy and Riley, 1962). For p-nitrophenyl phosphate, the enzymatic activity was

measured at 412 nm which is the absorbance range of p-nitrophenol (Hayes et al., 1999). The concentration of P or p-nitrophenol was determined as the subtraction of sample concentration and blank of reaction concentration Phytase activity with phytic acid and glyceraldehyde-3-phosphate as substrates was measured as P release measured by the Murphy-Riley method (Murphy and Riley, 1962).

To estimate $V_{max}$ and $K_m$, 200 mg of each phytase were suspended for 1h in solutions containing 0, 6.25, 12.5, 25, 50, 100 mM of P using the three substrates mentioned in the previous section (phytic acid, glyceraldehyde-3-phosphate and p-nitrophenyl phosphate). The reaction was stopped by the addition of 10% TCA. The kinetic parameters were determined by the graphical method of Lineweaver-Burk.

### 2.4 Statistical analysis

In order to find the pH and temperature value at which phytases show the maximum activity, different peak functions were adjusted with 2D Table Curve demo version. Experimental data of enzyme activity at different pH or temperatures were expressed as percentage of P released from each substrate and fitted to Lorentzian peak model for each treatment calculated following Eq. (1):

$$\%\ P\ released = \frac{a}{1+(\frac{x-b}{c})^2}, \quad (1)$$

Where a is the maximum percentage of P released; b is the pH value where the enzyme has maximum activity (a P release peak); c estimates the standard deviation of the distribution and x is the pH value. Parameters of each Lorentzian distribution for each enzyme and substrate were compared using F tests (Mead et al., 1993). In those cases where non-significant differences between enzymes (analyzed by F tests, analyzed by Statistix 9, student version) were found, a unified curve was fitted. The parameters and the obtained functions were compared by t-tests. Results obtained from the experiments of phytase distribution between soil solid and liquid phases were expressed as enzyme activity per soil gram (nkat g soil$^{-1}$). Exponential decay equations for enzyme distribution in liquid phase were fitted according to the Eq. (2):

$$y = (y_0 - b) * be^{-kx}, \quad (2)$$

where $y_0$ is the minimum enzyme activity in soil liquid phase, k is the relative exchange rate between the liquid phase and the solid phase and x is the time considered.

Exponential increase equations for enzyme distribution in the solid phase were fitted according to the Eq. (3):

$$y = y_{max} * (1 - e^{-kx}), \quad (3)$$

where $y_{max}$ is the maximum enzymatic activity in the solid phase of the soil, k is the relative exchange rate between the liquid phase and the solid phase and x is the reaction time. All functions where fitted by Table Curve 2D software. In those cases where significant differences between enzymes (analyzed by F tests) were not found, a unique curve was fitted. To determine the soil property effect on enzyme adsorption, the distribution of the enzymes between the solid and liquid soil phases were adjusted with linear functions between the enzyme activity and each analyzed soil property (Table 1).

## 3 Results and discussion

### 3.1 Phytase adsorption on soils

Figure 1 shows the distribution of phytases between liquid and solid phases in seven different soils of the Pampean Region (Mollisol order, Table 1). *A. niger* 1 showed the lowest adsorption to the solid phase, around 19% of the original substrate P content (Fig. 1e). This value remained stable after 30 minutes of incubation. *A. niger* 2 showed the greatest adsorption to the solid phase (40%, at 10 min Fig. 1f). *E. coli* 1 (Fig. 1g) presented 39% of binding to solid phase at 60 minutes whereas *E. coli* 2 presented a 37% binding to the soil solid phase at 5 minutes (Fig. 1h). This early maximum fixation prevented the fitting of a consistent function.

No linear relationship was observed between the parameter k and the analyzed soil characteristics for any of the four enzymes. In the case of $y_{max}$, we observed no linear relationship between soil characteristics for *A. niger* 1, 2 and *E. coli* 2. For *E. coli* 1, we found a significant correlation between the calcium content and $y_{max}$ (data not shown). Our results contrast with those reported by Yang and Chen (2017), who observed that soils showed a great variation in their capacity to retain phytases to the solid phase of the soil (19-40% observed in our work vs 17-93% in Yang and Chen (2017) work) and that sandy soils had the lowest phytase fixation. The soils used in this work did not have a wide range of texture. There is a tradeoff between phytase retention to the soil matrix and phytase activity, whose outcome would determine the real contribution of the enzyme to soil P availability. A low retention of phytases implies more enzyme in the soil solution and eventually a faster release of soil organic P. On the other side, phytases in soil solution could be denatured by soil microorganisms (Yang and Chen, 2017), whereas retained phytases would be released gradually, providing additional available P at later stages (Mezeli et al., 2017).

### 3.2 Biochemical characterization

Protein analysis indicated that *A. niger* 1, *A. niger* 2, *E. coli* 1 and *E. coli* 2 phytases had 4.2, 5.4, 8.2 and 2, 13.01 μg enzyme per mg of product, respectively. All four enzymes were effective at releasing inorganic P from the three offered organic P source. The four phytases released inorganic P from phytic acid along the whole range of pH following functions from which optimum and suboptimal pH values could be identified (Fig. 2). In both pH and temperature experiments, no significant differences were observed between *A. niger* 1 and 2 in terms of released inorganic P, thus data from both were pooled for performing the analyses.

#### 3.2.1 Effect of pH on enzyme activity

All enzymes were effective to release P from phytic acid throughout the analyzed pH range. *A. niger* optimum activity was observed at pH 5.9, value slightly higher than those reported earlier (5-5.5) (Konietzny and Greiner, 2002; Menezes- Blackburn et al., 2015; Sariyska et al., 2005). A 37% release of the original P contained in the substrate was observed at the peak activity section of the pH range (Fig. 2a). In *E. coli* 1 and *E. coli* 2 phytases (Fig. 2b and c) the peak activity was observed at pH 5.5 and 4.7, with 30% and 24% release of the initial P, respectively. Optimum pH values for *E. coli* were in line with previous reports (4.5-5) (Konietzny and Greiner, 2002; Menezes-Blackburn et al., 2015). The maximum value of $P_i$ released differed between $P_o$ sources (see coefficient *a* of Table 2), while the optimum pH for enzyme activity only differed between *A. niger* 1 + 2 and *E. coli* 2 (coefficient b of Table 2).

Phytase activity with p-nitrophenyl phosphate as substrate was notoriously diminished at pH values higher than 7.8, probably because the hydrolysis of the substrate. The peak activity of *A. niger* phytases was verified at pH 6.2 (Fig. 2g), with 50% P release. For *E. coli* 1 the maximum release of P was 37% at pH 5.8 (Fig. 2h), whereas for *E. coli* 2

the values were 24% and pH 5.9 (Fig. 2i). The comparison of the functions for the four enzymes revealed that they only differed in the *a* coefficient (Table 2), which represents the maximum P release. We did not find previous reports determining the optimum pH for enzyme activity with p-nitrophenyl-phosphate as organic P source.

For the substrate glyceraldehyde-3-phosphate, P release sharply decreased at pH values higher than 6 in *A. niger* and *E. coli* 1 enzymes, and at pH values higher than 8 in E. coli 2. *A. niger*, *E. coli* 1 and *E. coli* 2 enzymes showed a peak of activity at pH 3.9, 4 and 6, with a P release of 42 % (Fig. 2d), 37% (Fig. 2e) and 24% (Fig 2f), respectively. No statistical differences were observed on fitted coefficients between *A. niger* 1 + 2 and *E. coli* 1 functions, but these coefficients differed with the ones found for *E. coli* 2, revealing the particular shape of the function (Fig. 2i) (coefficients *a*, *b* and *c*, Table 2). As mentioned for p-nitrophenyl-phosphate, we did not find previous reports determining the optimum pH for enzyme activity with glyceraldehyde-3-phosphate as organic P source.

### 3.2.2 Effect of temperature on enzyme activity

The four enzymes remained active and could release Pi from the offered substrates throughout the whole temperature range evaluated (Fig. 3). When the substrate was phytic acid, both species of *A. niger* (1+2) showed the same response to temperature and consequently their functions were unified. The same occurred with *E. coli* 1 and 2 phytases. *A. niger* showed maximum activity at 24 °C (Fig. 3a), releasing 33% of the original P contained in the substrate. For *E. coli* enzymes (Fig. 3b), the peak was detected at 29 °C, with a 25% P release. The three coefficients of the function fitted for each pair of enzymes showed significant differences (Table 3), which reflects that *A. niger* had maximum release of P, but at a lower temperature than in *E. coli*. No difference between the four tested enzymes was found in the amount of P released. Obtained data on enzyme activity with phytic acid as substate agrees with Hayes et al. (1999), who found maximum activities in the 0-40ºC range. Other authors (Azeem et al., 2014; Sariyska et al., 2005) found maximum activities between 55ºC and 65ºC.

When the substrate was p-nitrophenyl-phosphate, the four enzymes showed a somewhat equivalent range of optimum temperatures than those found for phytic acid. For this substrate, the two *A. niger* enzymes showed the peak activity at 29 °C, releasing 17% of the substrate P (Fig. 3f). *E. coli* 1 phytase (Fig. 3g) released 22% of P at 29 °C and *E. coli 2* (Fig. 3h) also had the peak activity at 29 °C but lower P release: 13%. When comparing the coefficients of the fitted curves, they only differed in a coefficient (Table 3), representing the maximum P released.

When glyceraldehyde-3-phosphate was the substrate, the two *A. niger* enzymes had a similar behavior (Fig. 3c) with a peak activity at 24 °C and 10% release of the P contained in the substrate. *E. coli* 1 enzyme released 7% of the substrate P at 30 °C (Fig. 3d) and *E.coli* 2 (Fig. 3e) showed maximum activity at 20 °C, releasing 13% of the original P. No difference between adjusted coefficients of *A. niger* 1+2 and *E. coli* 1 functions was observed, but they differed with *E. coli* 2 coefficients (coefficients a, b and c in Table 3). We did not find previous reports determining the optimum temperature for phytase activity with p-nitrophenyl-phosphate and glyceraldehyde-3-phosphate as organic P sources.

### 3.2.3 Kinetic parameters

The response of the four enzymes to increasing concentrations of phytic acid is shown in Fig. 4 a-d. *A. niger* 2, *E. coli* 1 and 2 did not differ in the $V_{max}$ value (0.7 nkat mg$^{-1}$), while *A. niger* 1 showed a slightly lower value (0.6 nkat mg$^{-1}$). $K_m$ values of the four enzymes covered a narrow range (48 mM to 59 mM). *A. niger* 1 had the highest affinity (48.2 mM) followed by *E. coli* 1 (50.4 mM), *E. coli* 2 (54.3 mM) and *A. niger* 2 (59.2 mM). The $K_m$ values for phytic acid observed in our experiments were somewhat lower than those found by Konietzny and Greiner (2002) and

Menezes-Blackburn et al., (2015). These differences could be related to the methodological approach, i.e. buffer and temperature conditions. For example, some inhibitory effects of the $Ca^{2+}$ concentration of the buffer on the enzyme activity can affect the kinetic parameters (Vohra and Satyanarayana, 2003; Nannipieri et al., 2012). However, despite the relatively low enzyme affinity for phytic acid, the proportion of P released at optimum conditions was high (24% to 41% in one hour of incubation, Fig. 2).

Phytase activity of the purified enzymes in response to increasing concentrations of p-nitrophenyl phosphate showed a very narrow range of $V_{max}$ values (0.2 to 0.4 nkat $mg^{-1}$) (Fig. 4). *E.coli* 2 had the lowest $V_{max}$ and the highest substrate affinity (0.2 nkat $mg^{-1}$ and 22.8 mM), *E. coli* 1 (0.2 nkat $mg^{-1}$ and 25.8 mM), then by *A. niger* 1 (0.4 nkat $mg^{-1}$ and 51.7 mM) and finally *A. niger* 2 (0.4 nkat $mg^{-1}$ and 66.7 mM). These $K_m$ values are higher than those found by Soni et al. (2010) for *A. niger* phytases.

Finally, when the substrate was glyceraldehyde-3-phosphate (Fig. 4 e-h), a wide range of $V_{max}$ (4.2-60.7 nkat $mg^{-1}$) was observed for the four enzymes. *A. niger* 1 showed the lowest value (4.2 nkat $mg^{-1}$), followed by *A. niger* 2 (12.1 nkat $mg^{-1}$), *E. coli 2* (14.3 nkat $mg^{-1}$) and *E. coli* 1 (60.7 nkat $mg^{-1}$). $K_m$ values of the enzymes also had a wide range (2.4 mM to 34.1 mM). *A. niger* 1 showed the the highest affinity for this substrate (2.5 mM) followed by *E. coli* 2 (4.6 mM), *A. niger* 2 (5.2 mM) and *E. coli* 1 (34.1 mM). We did not find reports in the literature where the kinetic parameters of phytases were evaluated using glyceraldehyde-3-phosphate as substrate.

## 4 Conclusions

The prospects of using phytases as biofertilizers were evaluated in experiments performed under controlled conditions. The maximum enzyme activities were observed at pH values ranging from 3.9 to 6.2. All studied phytases remained active at the optimum soil pH range of the most productive agricultural soils. Optimal temperatures for phytase activity were also within the temperature range more suitable for most agricultural crops (20-30ºC). After being added to the soil, tested phytases showed a low adsorption to soil solid phase (20-40%). Phytases that remain in the solution could release Pi from the organic P of the soil, whereas phytases that remain adsorbed to the soil solid phase could be released later, providing an additional release of P. Our results suggest that purified phytases may constitute a feasible tool to be used as a complement to P fertilization. In such sense, further experiments should be performed to evaluate the enzyme performance under field conditions to evaluate the ability of phytases to release from organic soil P sources, their interaction with soil microorganisms and to test if crops can capitalize the eventual provision of inorganic P released.

## Author contributions

M. M. Caffaro, K. Balestrasse and G. Rubio designed the experiments and analyzed the data. M. M. Caffaro performed the experiments and analyzed the data with G. Rubio. Finally, M. M. Caffaro prepared the manuscript with the contribution of all co-authors.

## Conflicts of interest

Authors declare no conflict of interest regarding this research.

## Acknowledgments

Financial support was provided by CONICET, University of Buenos Aires and ANPCyT.

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

Tables

Table 1. Characteristics of the seven soils (0-20 cm) used in the phytase adsorption experiment.

| Soil | | Alberti | Adelia María | Lincoln | Oliveros | San Antonio de Areco | Balcarce | Balcarce |
|---|---|---|---|---|---|---|---|---|
| Soil type | | Typic Argiudoll | Entic Haplustoll | Typic Argiudoll | Typic Argiudoll | Typic Argiudoll | Typic Argiudoll | Typic Argiudoll |
| pH | | 5.9 | 6.3 | 6.0 | 5.7 | 6.1 | 6.5 | 5.9 |
| Ca | cmol$_c$ kg$^{-1}$ | 3.6 | 3.0 | 3.0 | 2.5 | 9.1 | 6.5 | 5.2 |
| Ca + Mg | | 4.5 | 4.0 | 4.0 | 3.2 | 6.1 | 7.1 | 5.6 |
| Ctotal | g kg$^{-1}$ | 26.0 | 11.5 | 14.2 | 14.0 | 20.2 | 38.6 | 36.9 |
| Clay | % | 16.3 | 16.7 | 8.8 | 28.8 | 30.0 | 27.6 | 36.4 |
| Sand | | 44.0 | 51.3 | 68.0 | 8.3 | 19.4 | 34.6 | 23.8 |
| Silt | | 39.8 | 32.0 | 23.3 | 63.0 | 50.6 | 36.5 | 48.6 |
| $P_{Bray\ 1}$ | mg kg$^{-1}$ | 14.9 | 16.2 | 3.4 | 14.9 | 3.4 | 24.6 | 35.6 |
| $P_{Mehlich\ 3}$ | | 20.3 | 19.3 | 12.9 | 20.8 | 6.9 | 36.1 | 48.6 |
| Ptotal | | 351 | 308 | 284 | 290 | 228 | 441 | 453 |
| Porg | | 208 | 148 | 150 | 181 | 163 | 339 | 325 |
| Pinorg | | 142 | 159 | 134 | 109 | 64 | 102 | 129 |
| Al | mmol$_c$ kg$^{-1}$ | 1.0 | 0.7 | 0.5 | 0.7 | 0.8 | 1.3 | 1.8 |
| Fe | | 1.3 | 1.1 | 1.3 | 1.1 | 1.4 | 1.9 | 2.3 |
| $Clay_{ssa\text{-}BET}$ | m$^2$ g$^{-1}$ | 12.6 | 9.8 | 3.5 | 13.7 | 31.4 | 20.5 | 32.5 |

Table 2. Coefficients of the adjusted functions for phytase activity (see graphs in Fig. 2) measured at different pH levels with phytic acid, p-nitrophenyl-phosphate and glyceraldehyde-3-phosphate as substrates. Four purified phytases (two isolated from *A. niger* and two from *E. coli*) were evaluated. In those cases where significant differences between enzymes (analyzed by F tests) were not found, a unique curve were fitted. Different letters correspond to significant differences between treatments (P <0.05, LSD procedure)

Adjusted function: $y = \frac{a}{1+(\frac{x-b}{c})^2}$

| Enzyme | Coefficients | | | $R^2$ |
|---|---|---|---|---|
| Substrate: Phytic acid | | | | |
| | a | b | c | |
| *A. niger* 1 + *A. niger* 2 | 36.6a | 5.9a | 2.7a | 0.73 |
| *E. coli* 1 | 30.1b | 5.5ab | 4.2a | 0.55 |
| *E. coli* 2 | 24.2c | 4.7b | 3.8a | 0.66 |
| Substrate: p-Nitrophenyl phosphate | | | | |
| *A. niger* 1 + *A. niger* 2 | 49.96a | 6.2a | 1.03a | 0.79 |
| *E. coli* 1 | 36.88b | 5.8a | 1.96a | 0.70 |
| *E. coli* 2 | 24.16c | 6.0a | 1.54a | 0.77 |
| Substrate: Glyceraldehyde-3-phosphate | | | | |
| *A. niger 1 + A. niger 2* | 44a | 3.9b | 0.7b | 0.94 |
| *E. coli 1* | 36.6b | 4.1b | 0.8b | 0.89 |
| *E. coli 2* | 24.2c | 6.0a | 1.5a | 0.77 |

Table 3. Coefficients of the adjusted functions for phytase activity (see graphs in Fig. 3) measured at different temperature levels with phytic acid, p-nitrophenyl-phosphate and glyceraldehyde-3-phosphate as substrates. Four purified phytases (two isolated from *A. niger* and two from *E. coli*) were evaluated. In those cases where significant differences between enzymes (analyzed by F tests) were not found, a unique curve were fitted. Different letters correspond to significant differences between treatments (P <0.05, LSD procedure).

Adjusted function: $y = \frac{a}{1+(\frac{x-b}{c})^2}$

| Enzyme | Coefficients | | | $R^2$ |
|---|---|---|---|---|
| Substrate: Phytic acid | | | | |
| | a | b | c | |
| *A. niger* 1 + *A. niger* 2 | 33.47a | 24a | 13.12b | 0.94 |
| *E. coli* 1+ E. coli 2 | 24.53b | 29a | 21.61a | 0.86 |
| Substrate: p-Nitrophenyl phosphate | | | | |
| *A. niger* 1 + *A. niger* 2 | 17.74b | 29a | 20.78a | 0.97 |
| *E. coli* 1 | 22.18a | 29a | 19.49a | 0.96 |
| *E. coli* 2 | 13.22c | 29a | 19.5a | 0.95 |
| Substrate: Glyceraldehyde-3-phosphate | | | | |
| *A. niger 1* + *A. niger 2* | 10.05a | 24b | 42.03b | 0.80 |
| *E. coli 1* | 6.62a | 30b | 36.34b | 0.84 |
| *E. coli 2* | 12.61b | 20a | 53.4a | 0.43 |

Figures

FIGURE 1. Phytase activity distributed in the liquid and solid phases for the phytase soil adsorption experiment. Four purified phytases (two isolated from *A. niger* and two from *E. coli*) were evaluated. Experiments were performed with the seven soils described in Table 1. For *A. niger* 1 and 2 and *E. coli* 1 phytases, a unique curve involving the seven soils was fitted because no significant differences (after F tests) were found between them. For *E. coli* 2, no function could be adjusted because a 37% binding to the soil solid phase was observed at 5 minutes and remained stable throughout the incubation period. Bars represent standard error of the mean.

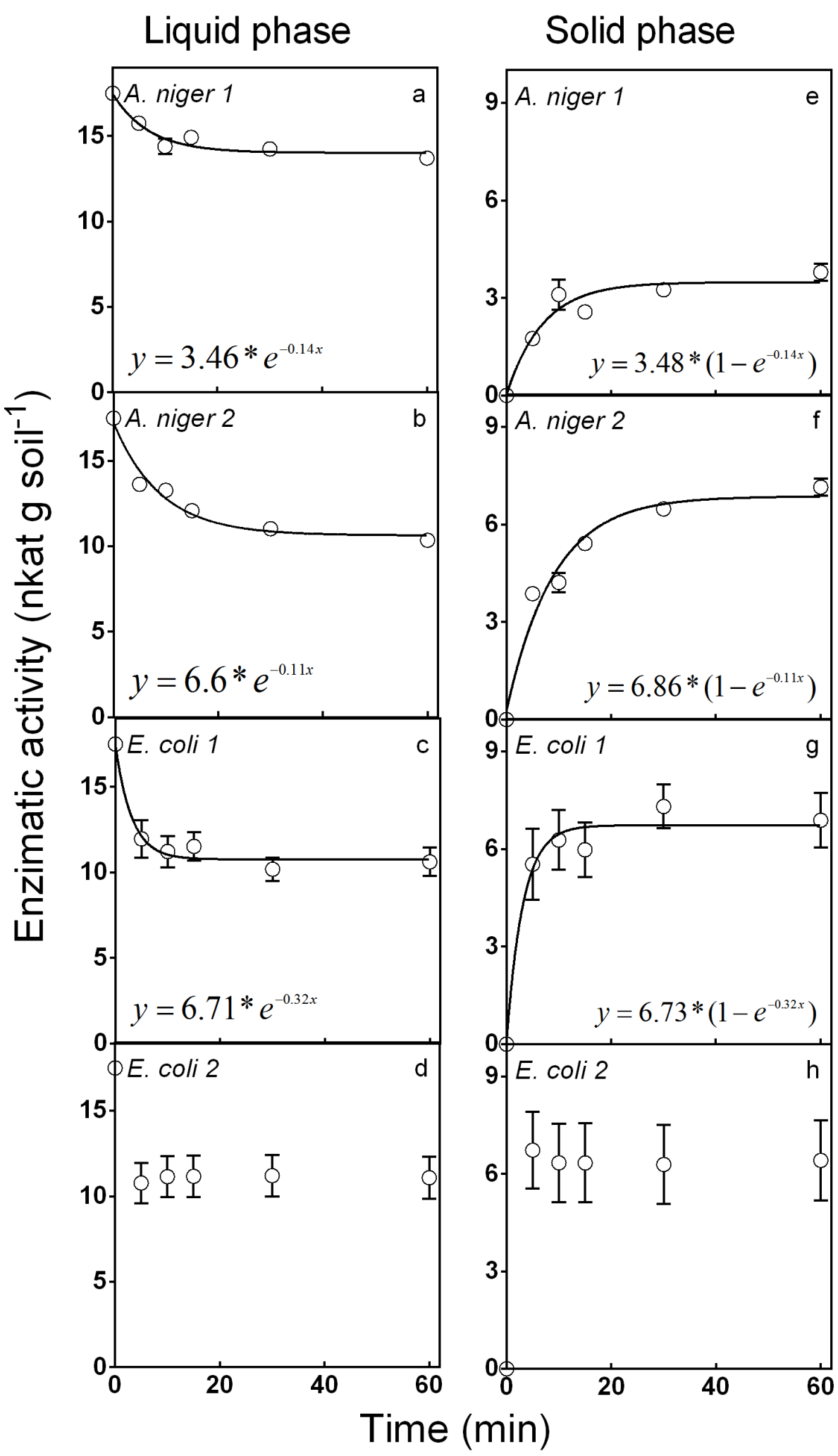

FIGURE 2. Phytase activity measured at different pH levels with phytic acid, p-nitrophenyl-phosphate and glyceraldehyde-3-phosphate as substrates. Four purified phytases (two isolated from *A. niger* and two from *E. coli*) were evaluated. In those cases where significant differences between enzymes (analyzed by F tests) were not found, a unique curve was fitted. Bars represent standard error of the mean. Coefficients of each adjusted model are observed in table 2.

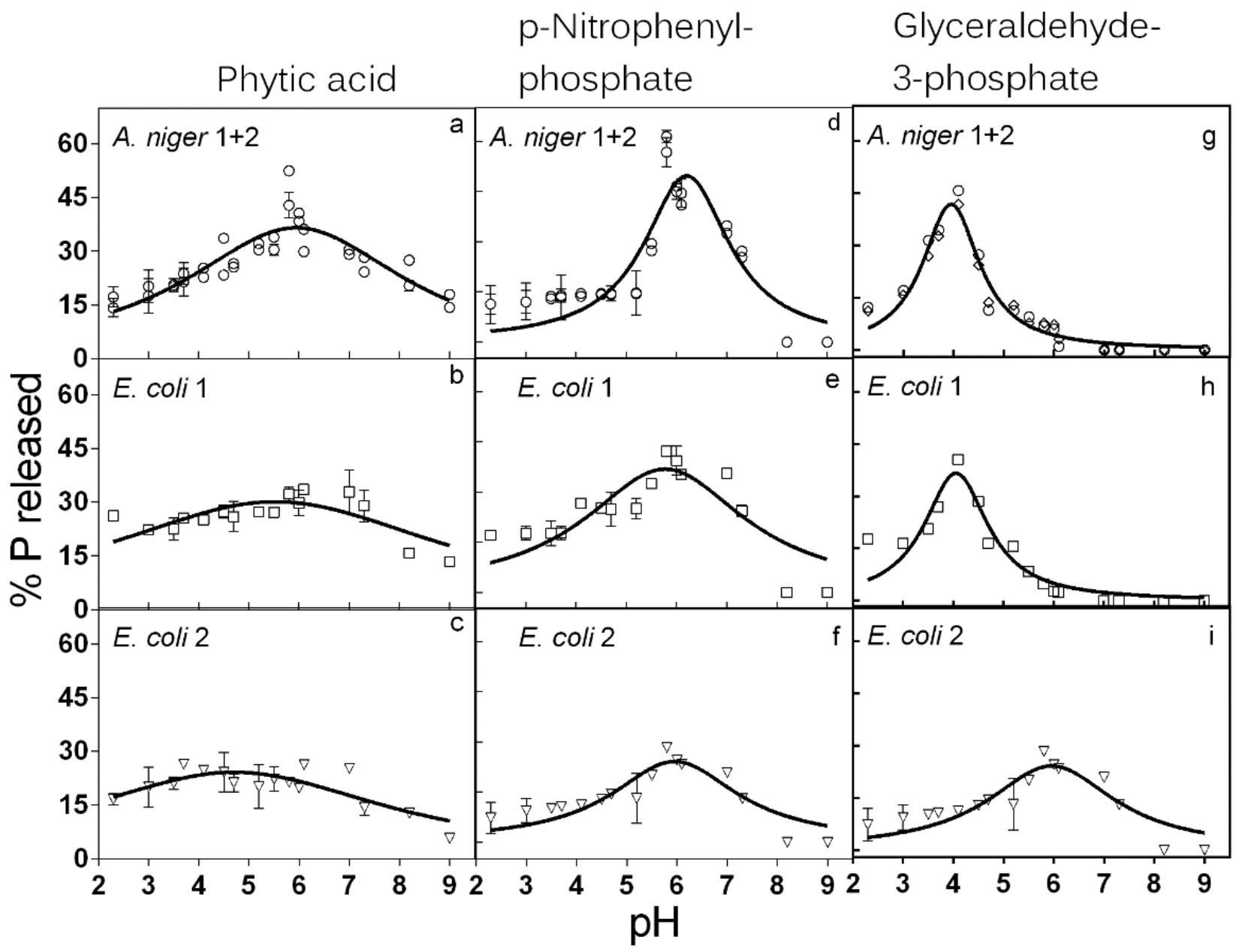

FIGURE 3. Phytase activity measured at different temperature levels with phytic acid, p-nitrophenyl-phosphate and glyceraldehyde-3-phosphate as substrates. Four purified phytases (two isolated from *A. niger* and two from *E. coli*) were evaluated. In those cases where significant differences between enzymes (analyzed by F tests) were not found, a unique curve was fitted. Bars represent standard error of the mean. Coefficients of each adjusted model are observed in table 3.

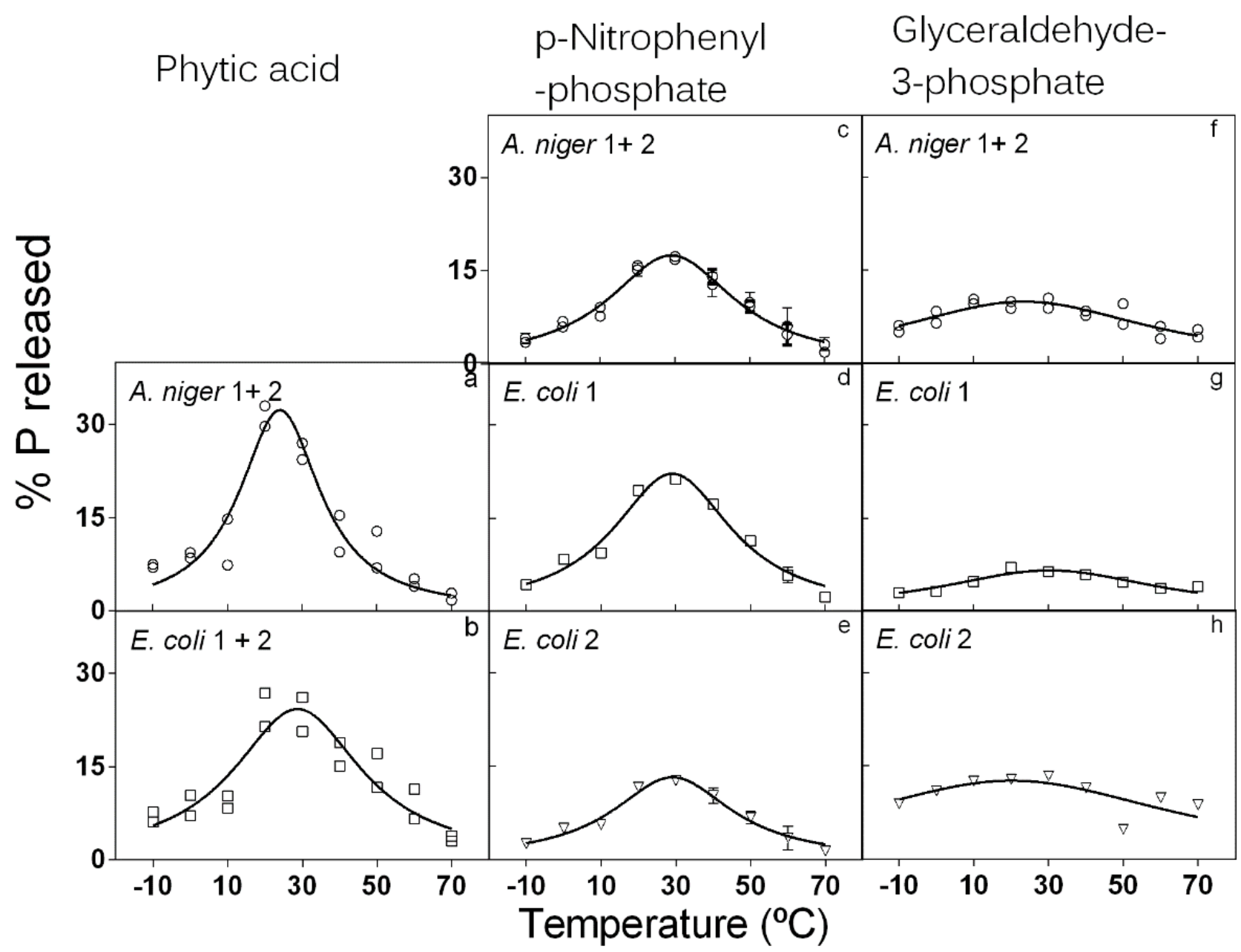

FIGURE 4. Kinetic parameters for phytic acid, p-nitrophenyl-phosphate and glyceraldehyde-3-phosphate as substrates of purified phytases (two isolated from *A. niger* and two from *E. coli*). The activity was determined at different concentrations of P contained in phytic acid as substrate. Bars represent standard error of the mean. Data were fitted to a Michaelis-Menten curve and the estimated $V_{max}$ and $K_m$ values obtained by the Lineaweaver-Burk method are shown.

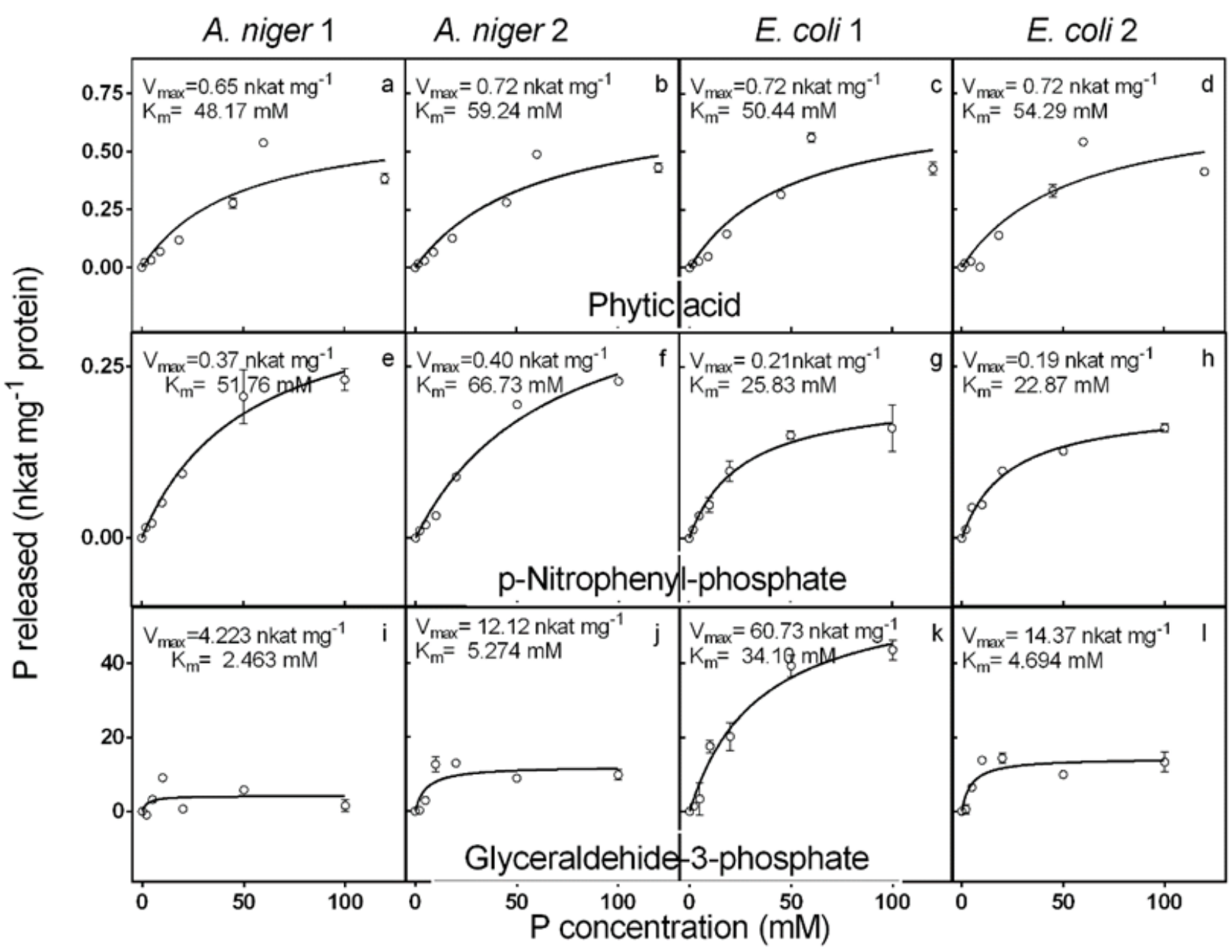