# Peer review of "Adsorption to soils and biochemical characterization of purified phytases"

_SOIL, 2019_

## Referee Comment (RC1) · Anonymous Referee #1 · 12 Oct 2019

Comments for editor The research work carried out under the theme "Adsorption to soils and biochemical characterization of purified phytases" is of scientific significance and has practical application for release of Pi from native or exogenously added organic P. Though, the study conducted is well organized but certain points need due attention. The Accession no of microbial strains used in the study is missing. The cost incurred on purchase of purified phytases and their availability needs mention. A comparative study with crude phytase obtained from wild strains of A. niger and E. coli could have also been conducted along side. The presentation is not up to the mark and needs modification. Some portions can be deleted and rewritten to make the publication more effective.

Technical Comments for authors 1. Abstract needs some modification indicating the

[Figure]

% increase in P release with A. niger over E. coli. 2. L-13-14 Please shift substrates pNP, G3phosphate and phytic acid after substrates 3. L-16 Please write that the order of P release from different substrates by A. niger and E. coli followed this trend (mention the trend) Introduction 4. L-24 Delete appropriate 5. There are approximately 38 references in introduction. The no. can be reduced. 6. L-45-48. The first phytase was discovered—— delete this paragraph. 7. L-74 instead of level write pH and temperature optima

Material and methods 8. L-77 A. niger in italics 9. L-79 powder form and not format 10. L-81 Superscript g-1 11. L-87 If one g soil was mixed with 20 ml phytase solution how can you take a sub sample of 500 ml. Please check the unit 12. L-92 150 ml or 150 microliter 13. L-105 total protein (Lowry et al ) 14. Phytase activity was measured with 3 substrates 15. L-119 Blanks for measuring enzyme activity included (i) (ii) (iii) 16. L123-126 Please rewrite this portion 17. 17. Please mention the amount of TCA added to stop the reaction 18. L-192 Modify the sentence 19. L-200 pH 7.8 was detrimental for release of Pi from pNP by A.niger 20. L-216 Change offered to tested substrates Results and discussion 21. Discussion part is totally missing and needs to be written properly 22. No explanation for findings is given Conclusion needs to rewritten 23. Tables and Fig titles need to be precise 24. Table 1 Mg+2 and Ca +2 and not Ca+1 25. Provide space between C total, P total and P inorganic 26. Fig.1. spelling for enzymatic

---

## Author Comment (AC1) · 26 Nov 2019

20-Nov-2019

Ref: MS No.: soil-2019-50. Title: Adsorption to soils and biochemical characterization of purified phytases Author(s): Maria Marta Caffaro et al.

Dear Executive Editors Soil Journal

We would like thank you and the anonymous reviewer for the valuable suggestions which helped us to greatly improve our ms. We have followed each one of the suggestions and the detailed responses and changes made to the original manuscript are given below and included in the new manuscript. If additional modifications are required, please let us know.

[Figure]

We are uploading a Word file with all the modifications included.

Regards,

Dr. Gerardo Rubio School of Agriculture - University of Buenos Aires

. . . . . . . . . . . . . . . . . . . . . . . . . . . . . . . . . . . . . . . . . . . ..

1. Comments for editor. The research work carried out under the theme "Adsorption to soils and biochemical characterization of purified phytases" is of scientific significance and has practical application for release of Pi from native or exogenously added organic P. Though, the study conducted is well organized but certain points need due attention. The Accession no of microbial strains used in the study is missing.

R: Thanks for your comment; in our experiment we used four commercial phytases. Two came from two different batches of A. niger commercially sold under the name "Habio phytases", which were obtained from Sichuan Habio Bioengineering Co.Ltd (Sichuan, China). The other two, came from two strains of E. coli. One is sold under the name "TS Smizyme phytases", obtained from Quimtia EDF (Buenos Aires Argentina) and the other is sold under the name "Ronozyme", obtained from DSM Nutritional Products Argentina S.A. Unfortunately, and as usual for commercial strains, no accession number was provided by the supplier. Anyway, we rearranged the paragraph to provide all available information.

Materials and methods now read as (New ms lines 68-73):

"Two phytases isolated from A. niger and two from E. coli were used in our experiments. In the first case, here named A. niger 1 and 2, phytases came from two different batches of the fungus which are commercially sold under the name "Habio phytases" by Sichuan Habio Bioengineering Co.Ltd (Sichuan, China), In the E.coli group, the first selected enzyme (here called E. coli 1) is sold under the name "TS Smizyme phytase", by Quimtia EDF (Buenos Aires Argentina), and the second (here called E. coli 2) is sold under the name "Ronozyme", by DSM Nutritional Products Argentina S.A."

. . .. 2. The cost incurred on purchase of purified phytases and their availability needs mention.

R: Enzymes were provided for free by the different companies producing and / or importing enzymes in the country. This information is included in the new version.

Materials and methods now read as (New ms line 73):

"These enzymes are in powder format at a concentration of 5000 U g-1 and was provided free of charge by the companies that produce or import them."

. . .. 3. A comparative study with crude phytase obtained from wild strains of A. niger and E. coli could have also been conducted alongside.

R: This would be a good option for the next phase of this investigation. Although it was was not the main objective of the present study. Anyway, this comment is highly appreciated and will be taken into account in our next phase.

. . .. 4. Technical Comments for authors Abstract needs some modification indicating the % increase in P release with A. niger over E. coli.

R: Done. Abstract now read as (New ms line 12): "Phytases from A. niger showed a higher capacity to release P, than phytases from E. coli (+13% on average)."

. . .. 5. L-13-14 Please shift substrates pNP, G3phosphate and phytic acid after substrates

R: Done. Acronyms now included throughout the whole ms. Abstract now read as (New ms line 11): ". . . . . p-nitrophenyl-phosphate (pNP), glyceraldehyde-3-phosphate (G3Phospahte) and phytic acid."

. . . 6. L-16 Please write that the order of P release from different substrates by A. niger and E. coli followed this trend (mention the trend).

R. OK. Iif we understood the meaning of the comment, the order is mentioned a couple

of lines above.

Abstract now read as (New ms lines 13-15): "All phytases were active throughout the pH and temperature ranges for optimum crop production. The amount of P that A. niger phytases release at pH that is commonly found in agricultural soils (5.5-7) is as follows: pNP > phytic acid > G3Phosphate, whereas in E. coli phytases the order was pNP / phytic acid > G3phosphate".

. . . 7. Introduction L-24 Delete appropriate

R: OK. Introduction now read as (New ms line 20): "Most strategies for enhancing P nutrition of agricultural crops aim to maintain soils at the convenient P critical level so that yields. . ."

. . .. 8. There are approximately 38 references in introduction. The no. can be reduced.

R: OK. In the new version, the number of references was reduced to 22.

. . .. 9. L-45-48. The fi̧rst phytase was discovered—– delete this paragraph.

R: OK, deleted.

. . .. 10. L-74 instead of level write pH and temperature optima.

R: OK. Introduction now read as (New ms line 65): . . . " the two evaluated phytases differ in their optimum pH and temperature to reach their maximum activity .."

. . .. 11. Material and methods L-77 A. niger in italics L-79 powder form and not format L-81 Superscript g-1

R. OK, all these editing issues were arranged as suggested.

. . .. 12. L-87 If one g soil was mixed with 20 ml phytase solution how can you take a sub sample of 500 ml. Please check the unit

R: We apologize because it was our mistake. The subsample volume is 500 microliters (not ml). Arranged in the new version.

Materials and methods now read as (New ms line 84): "...sub-samples of soil slurry (500 ïA▪l) were taken for phytase activity measurements..."

.... 13. L-92 150 ml or 150 microliter

R: Correct, same as above, it is 150 microliter. Arranged in the new version.

Materials and methods now read as (New ms line 86): "An aliquot (150 ïA▪l) of the soil slurry was used to measure the enzyme activity..."

.... 14. L-105 total protein (Lowry et al)

R. OK. Materials and methods now read as (New ms line 100): "Biochemical characterization of the phytases included: total protein (Lowry et al., 1951)"

.... 15. Phytase activity was measured with 3 substrates

R: OK, arranged as suggested

Materials and methods now read as (New ms line 102): "Phytase activity was measured with 3 substrates containing..."

.... 16. L-119 Blanks for measuring enzyme activity included (i) (ii) (iii)

R: OK. Materials and methods now read as (New ms lines 112- 113): "The activities were tested against three blanks: (i) reaction buffer without enzyme or substrate; (ii) reaction buffer with enzyme without substrate; and (iii) reaction buffer without enzyme with substrate."

.... 17. L123-126 Please rewrite this portion

R: OK. Materials and methods now read as (New ms line 114):

"For the pNP substrate, the enzymatic activity was measured at 412 nm which is the absorbance value of p-nitrophenol (Hayes et al., 1999). The concentration of 3 substrates was determined as the concentration of the whole sample minus the concentration of the reaction blank."
Please check that the sentence: "Phytase activity with phytic acid and glyceraldehyde-3-phosphate as substrates was measured as P release measured by the 125 Murphy-Riley method (Murphy and Riley, 1962)" was eliminated because this procedure is provided in the previous sentence.

…. 18. Please mention the amount of TCA added to stop the reaction

R: OK. Materials and methods now read as (New ms line 91): "Reactions were stopped with an equal volume of 10% TCA (300 ïĄ∎l in soil slurry experiments and 700 ïĄ∎l in soil solution experiments)."

…. 19. L-192 Modify the sentence

R: OK. Results now read as (New ms line 180): "All four enzymes were effective in releasing P from phytic acid throughout the entire pH range analyzed"

…. 20. L-200 pH 7.8 was detrimental for release of Pi from pNP by A.niger

R: OK, arranged as suggested. Results now read as (New ms line 186): "pH 7.8 was detrimental for release of Pi from pNP by A.niger, probably because the hydrolysis of the substrate

…. 21. L-216 Change offered to tested substrates

R: OK. Results now read as (New ms line 202):

"A. niger showed maximum activity at 24 °C (Fig. 3a), releasing 33% of the original P contained in the substrate."

…. 22. Results and discussion Discussion part is totally missing and needs to be written properly

R: OK, results and discussion sections were now written separately so as not to create confusion for the reader.

…. 23. No explanation for findings is given Conclusion needs to rewritten.

R: OK, conclusion section was reworded accordingly. After reorganizing the ms and split the Results and Discussion section into to separate parts, the conclusion was incuded in the last paragraph of the discussion. In this paragraph the test of hypothesis is specifically considered

Conclusion now read as (New ms lines 281- 295:

"Obtained results partially support our first hypothesis since the selected phytases showed a great ability to release P from different organic P sources, but A. niger 1, 2 and E. coli 1 release more P from pNP than phytic acid while E coli 2 has no preference for any particular substrate. In contrast, our results did not support the second proposed hypothesis, since the retention of phytases by the soil solid phase did not have a clear association with the analyzed soil properties. In this regard, it must be taken into account that the seven selected soils belonged to the Mollisol order. After being added to the soil, tested phytases showed an adsorption to soil solid phase ranging from 20 to 40%. Those phytases that remain in the solution could release Pi from the organic P of the soil, whereas phytases that remain adsorbed to the soil solid phase could be released later. Regarding our third hypothesis, although the evaluated phytases exhibited some differences in their pH and temperature levels to reach their optimum activity, all studied phytases remained active at the optimum soil pH range of the most productive agricultural soils (5-7). In the same line, optimal temperatures for phytase activity were also within the temperature range more suitable for most agricultural crops (20-30°C). Our results suggest that purified phytases may constitute a feasible tool to be used as a complement to P fertilization. In such sense, further experiments should be performed to evaluate the enzyme performance under field conditions to evaluate the ability of phytases to release from organic soil P sources, their interaction with soil microorganisms and to test if crops can capitalize the eventual provision of inorganic P released."

. . .. 23.Tables and Fig titles need to be precise

R: OK, all titles were rewritten

Tables and figures now read as:

"Table 1. Characteristics of seven representative soils of the Argentina's Pampa Region used for testing phytases adsorption. Samples were taken at 0- 20 cm, air dried and sieved at 1 mm prior to the analysis.

Table 2. Coefficients of the adjusted Lorentizian-peak functions for phytase activity (see graphs in Fig. 2) at different pH levels with phytic acid, p-nitrophenyl-phosphate and glyceraldehyde-3-phosphate as substrates. The equations were adjusted from the observed results of the release of P from each substrate used. Four purified phytases (two isolated from A. niger and two from E. coli) were evaluated. In those cases where significant differences between enzymes (analyzed by F tests) were not found, a unique curve were fitted. Different letters correspond to significant differences between treatments (P <0.05, LSD procedure) Coefficient a is the maximum percentage of P released; b is the pH value where the enzyme has maximum activity (a P release peak); c estimates the standard deviation of the distribution and x is the pH value.

Table 3. Coefficients of the adjusted Lorentizian-peak functions for phytase activity (see graphs in Fig. 2) at different temperature levels with phytic acid, p-nitrophenyl-phosphate and glyceraldehyde-3-phosphate as substrates. The equations were adjusted from the observed results of the release of P from each substrate used. Four purified phytases (two isolated from A. niger and two from E. coli) were evaluated. In those cases where significant differences between enzymes (analyzed by F tests) were not found, a unique curve were fitted. Different letters correspond to significant differences between treatments (P <0.05, LSD procedure) Coefficient a is the maximum percentage of P released; b is the temperature value where the enzyme has maximum activity (a P release peak); c estimates the standard deviation of the distribution and x is the temperature value.

FIGURE 1. Phytase activity distributed in the liquid and solid phases for the phytase soil

adsorption experiment. Four purified phytases (two isolated from A. niger and two from E. coli) were evaluated. Experiments were performed with the seven soils described in Table 1. For A. niger 1 and 2 and E. coli 1 phytases, a unique curve decay (Eq. 2), and exponential increase (Eq. 3) involving the seven soils was fitted because no significant differences (after F tests) were found between them. For E. coli 2, no function could be adjusted because a 37% binding to the soil solid phase was observed at 5 minutes and remained stable throughout the incubation period. Each point represents the average of three observations. Bars represent standard error of the mean.

FIGURE 2. Phytase activity measured at different pH levels with phytic acid, pNP and G3Phosphate as substrates. Four purified phytases (two isolated from A. niger and two from E. coli) were evaluated. In those cases where significant differences between enzymes (analyzed by F tests) were not found, a unique curve was fitted. Each point represents the average of three observations. Bars represent standard error of the mean. Coefficients of each adjusted model are shown in Table 2.

FIGURE 3. Phytase activity measured at different temperature levels with phytic acid, pNP and G3Pphosphate as substrates. Four purified phytases (two isolated from A. niger and two from E. coli) were evaluated. In those cases where significant differences between enzymes (analyzed by F tests) were not found, a unique curve was fitted. Each point represents the average of three observations. Bars represent standard error of the mean. Coefficients of each adjusted model are shown in Table 3.

FIGURE 4. Kinetic parameters for phytic acid, pNP and G3Pphosphate as substrates of purified phytases (two isolated from A. niger and two from E. coli). The activity was determined at different P concentrations (0 to 100 mM) contained in each substrate. Each point represents the average of three observations. Bars represent standard error of the mean. Data were fitted to a Michaelis-Menten curve and the estimated Vmax and Km values obtained by the Lineaweaver-Burk method are shown."

.... 24. Table 1 Mg+2 and Ca+2 and not Ca+1 25. Provide space between C total, P

]

total and P inorganic 26. Fig.1. spelling for enzymatic

R: OK, all these editing issues were arranged as suggested.

Please also note the supplement to this comment:
https://www.soil-discuss.net/soil-2019-50/soil-2019-50-AC1-supplement.pdf

—————————————————————

[Figure]

**Supplement:**

**Adsorption to soils and biochemical characterization of purified phytases**

[revised manuscript text omitted]

**2 Materials and methods**

**2.1 Enzyme preparation**

Two phytases isolated from *A. niger* and two from *E. coli* were used in our experiments. In the first case, here named *A. niger* 1 and 2, phytases came from two different batches of the fungus which are commercially sold under the name "Habio phytases" by Sichuan Habio Bioengineering Co.Ltd (Sichuan, China), In the *E.coli* group, the first selected enzyme (here called *E. coli* 1) is sold under the name "TS Smizyme phytase", by Quimtia EDF (Buenos Aires Argentina), and the second (here called *E. coli* 2) is sold under the name "Ronozyme", by DSM Nutritional Products Argentina S.A.". 
[revised manuscript text omitted]
 Mollisols of the Pampean Region which main soil properties are shown in Table 1. *A. niger* 1 phytase showed the lowest adsorption to the solid phase: around 19%, of the original substrate P content (Fig. 1e). This value remained stable after 30 minutes of incubation. In contrast, *A. niger* 2 phytase showed the greatest adsorption to the solid phase (40%, at 10 min Fig. 1f). *E. coli* 1 phytase (Fig. 1g) presented 39% of adsorption to the solid phase after 60 minutes of incubation whereas *E. coli* 2 presented a 37%

adsorption to the soil solid phase at 5 minutes (Fig. 1h). In this case, the early maximum fixation prevented the fitting of a consistent function.

To determine which soil characteristics (Table 1) affected phytase distribution between soil solid and liquid phases, a linear regression and correlation analysis between the parameters of Eq (3) $y_{max}$ (maximum distribution of the enzyme in the soil solid phase) and k (rate at which distribution peaks) with soil characteristics were performed. We observed no linear relationship between the parameter k and the analyzed soil characteristics for any of the four enzymes. In the case of $y_{max}$, we observed no linear relationship between soil characteristics for *A. niger*. Regarding *E. coli*, we found a significant correlation between the calcium content and $y_{max}$ for *E. coli* 1.

**3.2 Biochemical characterization**

Protein analysis indicated that *A. niger* 1, *A. niger* 2, *E. coli* 1 and *E. coli* 2 phytases had 4.2, 5.4, 8.2 and 2, 13.01 µg enzyme per mg of product, respectively. All four enzymes were effective at releasing inorganic P from the three offered organic P sources. The four phytases released P from phytic acid in the whole range of pH following functions from which optimum and suboptimal pH values could be identified. In both pH and temperature experiments, no significant differences were observed in P released between *A. niger* 1 and 2, thus data from both were pooled for performing the analyses.

**3.2.1 Effect of pH on enzyme activity**

All four enzymes were effective in releasing P from phytic acid throughout the entire pH range analyzed. When phytic acid was evaluated as a P source, the peak activity of *A. niger* phytases 1 and 2 (Fig. 2a) was observed at pH 5.9, with a release of 37% of the original P contained in the substrate. In *E. coli* 1 and *E. coli* 2 phytases (Fig. 2b and c), the peak activity was observed at pH 5.5 and 4.7, with 30% and 24% release of the initial P, respectively. The maximum value of released P differed between sources (coefficient *a* of Table 2), while the optimum pH for enzyme activity only differed between *A. niger* 1 + 2 and *E. coli* 2 (coefficient *b* of Table 2).

pH 7.8 was detrimental for release of Pi from pNP by *A.niger*, probably because the hydrolysis of the substrate. The peak activity of phytases was verified (>50% P release) at pH 6.2 (>50% P release) (Fig. 2g), while the maximum release of P was 37% at pH 5.8 for *E. coli* 1 enzyme (Fig. 2h) and 24% at pH 5.9 for *E. coli* 2 (Fig. 2i). The comparison of the functions for the four enzymes revealed that they only differed in a coefficient (Table 2), which represents the maximum P release.

For G3Phosphate as substrate, P release sharply decreased at pH values higher than 6 in *A. niger* and *E. coli* 1 enzymes, and at pH values higher than 8 in *E. coli* 2. *A. niger*, *E. coli* 1 and *E. coli* 2 enzymes showed a peak of activity at pH 3.9, 4 and 6, with a P release of 42 % (Fig. 2d), 37% (Fig. 2e) and 24% (Fig 2f) respectively. No statistical differences were observed on fitted coefficients between *A. niger* 1 + 2 and *E. coli* 1 functions, but these coefficients differed with the ones found for *E. coli* 2, revealing the particular shape of the function (Fig. 2i) for this enzyme (coefficients *a*, *b* and *c*, Table 2).

**3.2.2 Effect of temperature on enzyme activity**

The four enzymes remained active and could release P from the offered substrates throughout the whole temperature range (Fig. 3). When the substrate was phytic acid, both species of *A. niger* (1 + 2) and *E. coli* (1 and 2) showed the same behavior and consequently their functions were unified, *A. niger* showed maximum activity at 24 °C (Fig. 3a), releasing 33% of the original P contained in the tested substrate. In *E. coli* enzymes (Fig. 3b), the peak was detected at 29 °C, with

a 25% P release. The three coefficients of the function fitted for each pair of enzymes showed significant differences (Table 3), which reflects that *A. niger* had maximum release of P, but at a lower temperature than in *E. coli*.

When the substrate was pNP, the two *A. niger* enzymes showed the peak activity at 29 °C, releasing 17% of the substrate P (Fig. 3f). *E. coli* 1 phytase (Fig. 3g) released 22% of P at 29 °C and *E. coli* 2 (Fig. 3h) also had the peak activity at 29 °C but lower P release: 13%. When comparing the coefficients of the fitted curves, they only differed in a coefficient (Table 3), representing the maximum P released.

When G3Phosphate was the substrate, the two *A. niger* enzymes had a similar behavior (Fig. 3c) with a peak activity at 24 °C and a 10% release of the P contained in the substrate. In contrast, *E. coli* 1 enzyme released 7% of the substrate P at 30 °C (Fig. 3d) and *E. coli* 2 (Fig. 3e) showed maximum activity at 20 °C, releasing 13% of the original P. No difference between adjusted coefficients of *A. niger* 1 + 2 and *E. coli* 1 functions was observed, but they differed with *E. coli* 2 coefficients (coefficients *a*, *b* and *c* in Table 3).

**3.3 Kinetic parameters**

The response of the four enzymes to increasing concentrations of phytic acid is shown in Fig. 4 a-d. *A. niger* 2, *E. coli* 1 and 2 did not differ in the $V_{max}$ value (0.7 nkat mg$^{-1}$), while *A. niger* 1 showed a slightly lower value (0.6 nkat mg$^{-1}$). K values of the four enzymes covered a narrow range (48mM to 59 mM). *A. niger* 1 had the highest affinity (48.2mM) followed by *E. coli* 1 (50.4mM), *E. coli* 2 (54.3mM) and *A. niger* 2 (59.2mM).

Phytase activity of the purified enzymes in response to increasing concentrations of pNP showed a very narrow range of $V_{max}$ values (0.2 to 0.4 nkat mg$^{-1}$) (Fig. 4 e-h). *E. coli* 2 had the lowest $V_{max}$ and the highest substrate affinity (0.2 nkat mg$^{-1}$ and 22.8 mM), *E. coli* 1 (0.2 nkat mg$^{-1}$ and 25.8 mM), then by *A. niger* 1 (0.4 nkat mg$^{-1}$ and 51.7 mM) and finally *A. niger* 2 (0.4 nkat mg$^{-1}$ and 66.7 mM).

When the substrate was G3Phosphate (Fig. 4 i-l), a wide range of $V_{max}$ (4.2-60.7 nkat mg-1) was observed for the four enzymes. *A. niger* 1 showed the lowest value (4.2 nkat mg$^{-1}$), followed by *A. niger* 2 (12.1 nkat mg$^{-1}$), *E. coli* 2 (14.3 nkat mg$^{-1}$) and *E. coli* 1 (60.7 nkat mg$^{-1}$). $K_m$ values of the enzymes also had a wide range (2.4 mM to 34.1 mM). *A. niger* 1 showed the highest affinity for this substrate (2.5 mM) followed by *E. coli* 2 (4.6 mM), *A. niger* 2 (5.2 mM) and *E. coli* 1 (34.1 mM).

**4 Discussion**

The prospects of using phytases as biofertilizers were evaluated in experiments performed under controlled conditions. Phytases are polar molecules with negative charge that can be retained by the soil matrix, affecting their capacity to mineralize organic sources of soil P (George et al., 2005; Yang and Chen, 2017). For example, George et al., (2005) observed a strong adsorption of *A. niger* phytases to the soil solid phase (57-86%), especially on clayey or acid soils with high P adsorption capacity. As an approximation to the use of phytases as a complement for plant P nutrition, we evaluated the distribution of phytases in the solid phase of seven agricultural soils Mollisols differing in texture and P adsorption capacity (Table 1). After 60 min of incubation, the proportion of phytases found in the solid phase was lower than in the liquid phase (23-34% vs. 66-77%, Fig. 1). Our results contrast with those reported by Yang and Chen (2017), who observed that soils showed a great variation in their capacity to retain phytases to the soil solid phase of the soil (19-40% observed in our work vs 17-93% in Yang and Chen (2017) work) and that sandy soils had the lowest phytase fixation. These differences may be due that our soils used in this work did not have had a narrower wide range of texture. The benefits of having a low adsorption to the soil matrix for phytases as potential biofertilizers are not as straightforward. There is a tradeoff between phytase retention to the soil matrix adsorption and phytase activity, whose outcome would

determine the real contribution of the enzyme to soil P availability. A low retention of phytases implies more enzyme in the soil solution and eventually a faster release of soil organic P. On the other side, phytases in soil solution could be more easily denatured by soil microorganisms (Yang and Chen, 2017), whereas retained phytases would be released gradually, providing additional available P at later stages (Mezeli et al., 2017).

The four enzymes were effective to release P from phytic acid throughout the analysed pH range. *A. niger* optimum activity was observed at pH 5.9, value slightly higher than those reported by earlier reports (5-5.5) (Konietzny and Greiner, 2002; Menezes- Blackburn et al., 2015; Sariyska et al., 2005). In *E. coli*, optimum pH was observed between 4.7 and 5.5, in agreement with values reported earlier (4.5-5) (Konietzny and Greiner, 2002; Menezes-Blackburn et al., 2015). We did not find previous reports determining the optimum pH for enzyme activity with pNP and G3Phosphate as organic P sources.

Several methodologies have been used to evaluate phytase activity along a temperature range. In some cases, only optimum temperatures were reported (Azeem et al., 2014), whereas other authors reported the release of inorganic P as relative maximum values (Hayes et al., 1999). In our experiments, we found an optimum range phytase activity ranging between 10ºC and to 40º C for phytic acid, releasing in one hour up to 30% of the P contained in the substrate. These data agree with Hayes et al. (1999), who found maximum activities in the 0/40ºC range, although other authors (Azeem et al., 2014; Sariyska et al., 2005) found maximum activities between 55ºC and 65ºC. When the substrate was pNP, the four enzymes showed a somewhat equivalent range of optimum temperatures than those found for phytic acid. However, the proportion of P released from pNP was almost half that observed from phytic acid, and never exceeded 20% (Fig. 3). The optimal enzyme temperature range for G3Phosphate was difficult to determine due to the scarce proportion of P released (5%). The fitted functions did not show a clear peak such as the ones observed for the other organic P sources (Fig. 3).

Enzyme kinetic analysis indicated that the affinity ($K_m$) for phytic acid of *A. niger* enzyme showed a range of 48-59 mM (Figs. 4a and 4b) values lower than those found by Konietzny and Greiner, (2002) and Menezes-Blackburn et al., (2015). In *E. coli*, the range of $K_m$ for phytic acid obtained in our experiments was 50-54 mM (Fig. 4), which indicates a lower affinity compared to the 130-630 µM range reported by Konietzny and Greiner (2002). For pNP as substrate, Soni et al. (2016) reported a Km range of *A. niger* phytases of 1 to 4 mM, values much lower than those found in our experiments (52-67 mM, Fig. 4). We did not find reports in the literature where the kinetic parameters of phytases were evaluated using G3Phosphate as substrate. We observed that A. niger 1 and E. coli 2 phytases have the highest affinity for G3Phosphate (2-4.7 mM, Fig 4).

The observed differences between values $K_m$ for phytic acid in our experiments compared with literature could be related to methodological differences, since there is no common protocol for evaluating purified phytases, for example the 
[revised manuscript text omitted]

470

[Figure]

FIGURE 4. Kinetic parameters for phytic acid, pNP and G3Pphosphate as substrates of purified phytases (two isolated from *A. niger* and two from *E. coli*).  The activity was determined at different P concentrations (0 to 100 mM) contained in each substrate. Each point represents the average of three observations. Bars represent standard error of the mean. Data were fitted to a Michaelis-Menten curve and the estimated $V_{max}$ and $K_m$ values obtained by the Lineaweaver-Burk method are shown.

[Figure]

---

## Referee Comment (RC2) · Anonymous Referee #2 · 10 Jan 2020

The paper adsorption to soils and biochemical characterization of purified phytases, by Caffaro et al, uses conventional techniques for the evaluation of known commercial phytases. They have some success trying to prove that phytases have the potential to be used as complement for soil fertilizers. There are many issues that need to be clarified before publication: The title itself is ambiguous and misleading Recently it has been a discussion about the term phytase. Certainly one definition is that all enzymes which area use phytate as substrate are phytases. However, several authors i.e Greiner have pointed out that many of those are actually phytate degrading enzymes particularly the ones in E coli. Therefore it might be that those are not true phytases, The main reason is that their function is not related to processing phytate, different from some other's "real" phytases in plants i.e. PAPphy. The authors refer to the work

of Misset 2003 as a reference of E coli phytases and their relevance in the industry. There are a couple of issues here. First I'm not really sure of the relevance of all strains of E coli phytases for the industry. If any which ones?. Many strains of E coli possess an active phosphatase A gene witch can provide a certain level of phytate degradation but a real level of commercial degradation, I'm not sure about it.

Only until the lines 60 to 70, the really important point of the work was revealed. The main point in my perspective is the usage of phytases as biological fertilizers to release inorganic P from organic P sources. But so far the whole history sounded more focused on something else. The major problem of the paper starts with the first hypothesis: Phytases have the ability to release P from different organic P sources, with a preference for phytic acid. In that way is redacted that is not a hypothesis that contributes at all with new knowledge in the field. It is already known that some phytases are highly specific and others are not but have preferences for phytate. Similar to the other two. Many references for that just two examples: doi:10.1128/AEM.01384-15 doi: 10.1128/mBio.01966-18 Is the norm of the journal to include only some of the line numbers? That makes it more difficult for review. The abstract is very misleading because implies that the authors isolated the phytases from the fungi by themselves. That is not the case. Line 18-19: The proportion of phytases found in the solid phase of the soil 60 minutes after addition was lower than that found in the liquid phase (23-34% vs. 66-77%). This result is not well connected in the abstract, is coming out of nowhere.

Lines 38-39: There are different forms of inositol-phosphates and the most abundant from phytate (refers only to the salt form). But what exactly is the meaning of phytates in these lines and in the subsequent text in general?.

Line 48. E coli and the rest of the text please italicize where required

The hypotheses are not real hypotheses in the way their current state. It is already known that phytases can use different substrates. The number two was proved by a paper that the authors cite https://doi.org/10.1002/jpln.201600421. Finally, the hypothesis number 3 is way too basic for being a good work hypothesis

The biochemical characterization needs to include the catalytic efficiency of the reactions. It has been demonstrated recently by the works of Tan et al (doi: 10.1007/s00253-015-7097-9) and others in 2019 using metagenomes that phytases are also present in metagenomes of soils. In fact, their presence is underestimated. Where is the experiment which proves that the used soils have low phytase activity?

The control reactions of the initial experiments are missing.

Line 134 I don't think is a good idea to use a demo or student versions of any software for statistical analysis in a publication.

The characterization of the enzymes are very poor. Soils are complex matrices with multiple variants i.e ions. That was not evaluated here.

Were the buffers set at the optimal temperature ?

The authors refer at the begging of the manuscript to the type of enzymes as 3-phytases. But they do not mention what type of enzymes are from the structural point of view. Are they acid phytases? Maybe that's is why need pH relatively low to act. But nothing of this is mentioned in the text.

Is the optimal pH was determined before that the temperature is obvious that they did not set the buffer for the pH test at the right temperature. Therefore the pH characterization is not trustworthy.

It seems that the authors did not review any literature about phytases in 2019.

---

## Author Comment (AC2) · 30 Jan 2020

Jan 29th 2020

Ref: MS No.: soil-2019-50. Title: Adsorption to soils and biochemical characterization of purified phytases Author(s): Maria Marta Caffaro et al.

Dear Executive Editors Soil Journal

We would like thank you and the anonymous reviewer (II) for the valuable suggestions which helped us to greatly improve our ms. We have followed each one of the suggestions and the detailed responses and changes made to the original manuscript are given below and included in the new manuscript. If additional modifications are required, please let us know.

We are uploading a Word file with all the modifications included. This modifications will be added to the response to reviewer I.

Regards,

Dr. Gerardo Rubio (corresponding author) School of Agriculture - University of Buenos Aires

. . . . . . . .. . . . .. . . . . .. . . . . .. . . .. Interactive comment on "Adsorption to soils and biochemical characterization of purified phytases" by Maria Marta Caffaro et al.

Anonymous Referee #2

1. The paper adsorption to soils and biochemical characterization of purified phytases, by Caffaro et al, uses conventional techniques for the evaluation of known commercial phytases. They have some success trying to prove that phytases have the potential to be used as complement for soil fertilizers. There are many issues that need to be clarified before publication: The title itself is ambiguous and misleading. Recently it has been a discussion about the term phytase. Certainly, one definition is that all enzymes which area use phytate as substrate are phytases. However, several authors i.e Greiner have pointed out that many of those are actually phytate degrading enzymes particularly the ones in E coli. Therefore it might be that those are not true phytases, The main reason is that their function is not related to processing phytate, different from some other's "real" phytases in plants i.e. PAP phy.

R: Yes, definitely, not all enzymes that are capable of degrading phytates are "real" phytases. According to Misset (2003), "real" phytases are those enzymes capable to degrade completely phytate molecules and to release all phosphates contained in them. However, the term phytases is a topic of debate as pointed out by the reviewer. Moreover, one paper of the author cited by the reviewer includes the specific and general term in the title (Konietzny, U., & Greiner, R. 2002. Molecular and catalytic properties of phytate‐degrading enzymes (phytases). Int. J. Food Sci. Tech., 37, 791-812." ).

Enzymes used in this work are commercially sold under the generic name "phytases". We did not perform tests to evaluate the three-dimensional structure of the enzyme because that issue was beyond the scope and objectives in this stage of our research. However, many authors refer the phytate degrading enzymes from E. coli as" phytases" (e.g. Menezes-Blackburn et al. 2011, doi:10.1016/j.biortech.2011.07.054; Derjsant-Li and Kwakernaak 2019, doi: 10.1016/j.anifeedsci.2019.05.018). Then, we believe that it is correct to use the term phytase to describe enzymes that degrade phytates from E. coli (and A. niger.) Taking into account this explanation, we enlarge the introduction to clarify this issue. Now read as:

"Phytases are enzymes released by bacteria, fungi, plants and animals (Jorquera et al., 2008) and are able to catalyze the release of P from phytates. Phytases have the ability to release the 6 Pi molecules that are contained in phytate (Misset 2003)."

Regarding the title, an alternative one may be: "Adsorption to soils and biochemical characterization of phytate-degrading enzymes (phytases)" (following Konietzny and Greiner 2002). We are ready to move for this title if the editor and reviewer consider this as a better option. Anyway, we propose to maintain the original title "Adsorption to soils and biochemical characterization of purified phytases", since "a priori" we did not know the real capacity of the purified enzymes to release P.

. . . 2. The authors refer to the work of Misset 2003 as a reference of E coli phytases and their relevance in the industry. There are a couple of issues here. First I'm not really sure of the relevance of all strains of E coli phytases for the industry. If any which ones?.

R: Commercial phytases available in Argentina are generally purified from A. niger and E.coli, so we considered necessary to cite previous reports about E.coli as a source of phytase. Our E coli enzymes came from commercialized products mainly used for animal feed application. The strains are called with the same name as the product (TS Smizyme phytase, by Quimtia EDF, and Ronozyme, by DSM Nutritional Products

Argentina S.A.). We used the same approach than Menezes-Blackburn et al. 2015 doi: 10.1021/acs.jafc.5b01996, who mention three E. coli strains that are isolated and commercialized. See below the table extracted from this paper as an example of how they describe the strain.

SEE TABLE IN THE SUPPLEMENT FILE "soil-2019-50-supplement.pdf"

.. 3. Many strains of E coli possess an active phosphatase A gene witch can provide a certain level of phytate degradation but a real level of commercial degradation, I'm not sure about it

R: We agree. That's why we decided to perform these experiments to verify the actual activity of commercial enzymes purified from E. coli and A. niger.

. . . 4. Only until the lines 60 to 70, the really important point of the work was revealed. The main point in my perspective is the usage of phytases as biological fertilizers to re-lease inorganic P from organic P sources. But so far the whole history sounded more focused on something else. The major problem of the paper starts with the first hypothesis: Phytases have the ability to release P from different organic P sources, with a preference for phytic acid. In that way is redacted that is not a hypothesis that contributes at all with new knowledge in the field. It is already known that some phytases are highly specific and others are not but have preferences for phytate. Similar to the other two. Many references for that just two examples: doi:10.1128/AEM.01384-15 doi:10.1128/mBio.01966-18

R: OK, we agree that our hypothesis can lead to misunderstandings. We are not talking about phytases in general, but specifically about the commercial enzymes of our work. First hypothesis now read as . . . "four commercially available phytase products tested in this work have the ability to release P from different organic P sources, with preference for phytic acid". See our reply to comment 7 for more explanations on this topic.

. . . 5. Is the norm of the journal to include only some of the line numbers? That makes

it more difficult for review.

R: Yes, we use the journal template for submitting the paper.

. . . 6. The abstract is very misleading because implies that the authors isolated the phytases from the fungi by themselves. That is not the case. Line 18-19: The proportion of phytases found in the solid phase of the soil 60 minutes after addition was lower than that found in the liquid phase (23-34% vs.66-77%). This result is not well connected in the abstract, is coming out of nowhere.

R: OK, we rearranged the paragraph. Abstract now read as . . ." Four purified phytases isolated from Aspergillus niger and Escherichia coli were characterized biochemically and in terms of their adsorption to soils belonging to the Mollisol order. Three different organic P substrates were used to measure enzyme activity in a wide range of pH (2.3 to 9) and temperatures (-10° to 70°C): p-nitrophenyl-phosphate (pNP), glyceraldehyde-3-phosphate (G3Phospahte) and phytic acid. Phytases had low affinity for the solid phase (23-34%of adsorption after one hour of incubation). Phytases from A. niger showed a higher capacity to release P, than phytases from E. coli (+13% on average). All phytases were active throughout the pH and temperature ranges for optimum crop production under field conditions. The amount of P that A. niger phytases release at pH values commonly found in agricultural soils (5.5-7) was as follows: pNP > phytic acid > G3Phosphate, whereas in E. coli phytases the order was pNP / phytic acid > G3phosphate. Obtained results are promising in terms of the use of phytases as a complement to P fertilization in agricultural settings and encourages further studies under field conditions.."

. . . 7. Lines 38-39: There are different forms of inositol-phosphates and the most abundant from phytate (refers only to the salt form). But what exactly is the meaning of phytates in these lines and in the subsequent text in general?.

R: In our work, we want to test the ability of commercial products to release P from different P organic sources, so in this paragraph we introduce the different forms of

organic P found in the soil and in what proportion they are found. Please take into account our reply to comment 1, in which we explain that the definition of phytases is enlarged in the new version.

... 8. Line 48. E coli and the rest of the text please italicize where required.

R: OK. Done

... 9. The hypotheses are not real hypotheses in the way their current state. It is already known that phytases can use different substrates. The number two was proved by a paper that the authors cite https://doi.org/10.1002/jpln.201600421. Finally, the hypothesis number 3 is way too basic for being a good work hypothesis.

R: We understand the point raised by the reviewer. The original hypotheses may appear as basic for ultra-purified enzymes or recombinant proteins produced for academic or related activities. Previous studies about phytases mainly come from ultrapurified enzymes like the ones provided by lab-supplies companies such as Sigma and used in academic labs. It is clear that this high quality but extremely expensive products cannot be used in real agricultural by farmers. In this report we tested at which extent commercial phytases have comparable performance than the ultrapurified enzymes. We think that it is not correct to extrapolate results from both type of products. In such sense, in the new version we will modify the text highlighting the commercial nature of our evaluated phytases.. Anyway, we will modify the hypotheses by clarifying that we refer to these commercial enzymes. Hypothesis now read as... " i) the four commercially available phytase products tested in this work have the ability to release P from different organic P sources, with preference for phytic acid, but differ in the pH and temperature levels to reach their optimum activity ii) the retention of commercial phytases in the soil solid phase is associated to the soil clay content.

... 10. The biochemical characterization needs to include the catalytic efficiency of the reactions.

R: Done. Information was added to fig. 4

SEE FIGURE IN THE SUPPLEMENT FILE "soil-2019-50-supplement.pdf"

11. It has been demonstrated recently by the works of Tan et al (doi:10.1007/s00253-015-7097-9) and others in 2019 using metagenomes that phytases are also present in metagenomes of soils. In fact, their presence is underestimated. Where is the experiment which proves that the used soils have low phytase activity?. The control reactions of the initial experiments are missing.

R: Very good observation. In all experiments, we use blank reactions to ensure that the results presented in this work are those observed by the interaction of the enzymes with the soil. We measure soil phytase activity without the addition of the enzyme and observe soil phytase activity less than 1nkat g soil -1. This topic was not clear enough in the previous version of the ms. The sentence in Materials and methods was reworded as "Phytase activity of the soil suspension was calculated as the difference between the soil suspension with enzyme minus the soil suspension without enzyme."

12. Line 134 I don't think is a good idea to use a demo or student versions of any software for statistical analysis in a publication.

R: Table Curve Demo version gives a limited period of software use (30 days), but it has the same mathematical functions as the full version. Anyway, we agree to remove the "demo version" as required. Statistical analysis was performed with the Statistix student version which is the software that we used in our lab since long time ago and was tested several times. For this particular paper and following this comment, we performed again all our analysis with INFOSTAT software (https://www.infostat.com.ar/) and the results were exactly the same. This software is cited in the new version.

. . . 13. Were the buffers set at the optimal temperature?

R: The buffers were prepared at normal room temperature (20-24 °C) and the incubation experiments were performed at 25 °C (for evaluating optimum pH and kinetic

parameters) and along a temperature range (-10-70°C) for evaluating optimum temperature for enzyme activlty. In this last experiment pH was set as 5.5. For these experiments we followed the approach proposed by George et al. (2005) and Hayes et al. (1999). The original text was somewhat unclear at this point and we reworded the sentence accordingly.

. . . 14. The authors refer at the begging of the manuscript to the type of enzymes as 3-phytases. But they do not mention what type of enzymes are from the structural point of view. Are they acid phytases? Maybe that's is why need pH relatively low to act. But nothing of this is mentioned in the text. Is the optimal pH was determined before that the temperature is obvious that they did not set the buffer for the pH test at the right temperature. Therefore the pH characterization is not trustworthy.

R: Commercial phytases used as a complement to poultry nutrition must be active at the stomach pH of the animals (about pH 3). Phytases that are active at that pH value are acid phytases, so in the introduction we mention the 3-phytases and 6- phytases that are by definition acid phytases (lines 41-50 of new paper version:"A. niger phytases are mainly extrinsic (Azeem et al., 2015), and are classified as 3-phytases, because they primarily dephosphorylate the phosphate group located at 3-position. E. coli phytases are mainly membrane-associated proteins and were classified as 6-phytase (Azeem et al., 2015). The classification as 3- or 6-phytases is related to which phosphate group is attacked first and would be determined by conformational differences in the -domain of each phytase (Konietzny and Greiner, 2002)"). The temperature during the pH experiment was set at 25 °C. This temperature is within the optimal range found for the 4 tested enzymes (20 to 29°C). On the other hand, 25 °C is also within the optimal range of crops growing under field conditions, which are the final context of our line of work on phytases in future experiments. These 25 °C are also clearly within the optimal range of the set of buffers used to generate the pH range (e.g. Hayes et al. 1999; cited in the ms).

. . . 15. It seems that the authors did not review any literature about phytases in 2019.

R: To the best of our knowledge, we checked all papers on phytases published in top journals before the submission date of our SOIL ms. We cited the most relevant papers in each section but probably we missed some of them.

Please also note the supplement to this comment:
https://www.soil-discuss.net/soil-2019-50/soil-2019-50-AC2-supplement.pdf

———————————————————

---

## Author Response (AR1)

February 20th, 2020

Ref: MS No.: soil-2019-50. Title: Adsorption to soils and biochemical characterization of purified phytases.Maria Marta Caffaro et al.

Dear Topical Editor Dr Jeanette Whitaker SOIL Journal

We would like you and the anonymous reviewers for their valuable suggestions which helped us to greatly improve our ms. We have followed each one of the suggestions and the detailed responses and changes made to the original manuscript are given below and included in the new manuscript. If additional modifications are required, please let us know.

We are uploading a Word file with all the modifications included and a second file with the annotated version.

Regards,

Dr. Gerardo Rubio School of Agriculture - University of Buenos Aires

.....

Detailed response to the Editor and the reviewers:

.....

**Topical Editor**

1. Thank you for your detailed responses to the reviewers comments. There are quite a number of substantive changes and improvements which the reviewers recommend to improve the clarity and presentation of the manuscript. If you would amend the manuscript as you have described in your responses then the manuscript will be reconsidered for publication. On one specific issue, reviewer 1 comment 23 recommends the table and figure titles need to be more precise. Your suggested changes result in titles which are too long and contain methodological information which does not need to be included. I would encourage the authors to construct specific titles which enable the reader to understand the table or figure without reference to the text but in a more concise way. Where a figure has related data in a table this can be simply referenced to the respective table.

R: We incorporated the comments made by the two reviewers into the new version of the manuscript. Legends of the tables and figures were rewritten to make them more concise.

.....

Anonymous Referee #2

1. The paper adsorption to soils and biochemical characterization of purified phytases, by Caffaro et al, uses conventional techniques for the evaluation of known commercial phytases. They have some success trying to prove that phytases have the potential to be used as complement for soil fertilizers. There are many issues that need to be clarified before publication: The title itself is ambiguous and misleading. Recently it has been a discussion about the term phytase. Certainly, one definition is that all enzymes which area use phytate as substrate are phytases. However, several authors i.e Greiner have pointed out that many of those are actually phytate degrading enzymes particularly the ones in E coli. Therefore it might be that those are not true phytases, The main reason is that their function is not related to processing phytate, different from some other's "real" phytases in plants i.e. PAP phy.

R: Yes, definitely, not all enzymes that are capable of degrading phytates are "real" phytases. According Misset (2003), the "real" phytases are those enzymes capable to degrade completely the phytate molecules and release all the phosphates contained in it. However, the term phytases is a topic of debate as pointed out by the reviewer and one paper of the author cited by the reviewer includes the specific and general term in the tile (" Molecular and catalytic properties of phytate-degrading enzymes (phytases)" ).

Enzymes used in this work are commercially sold under the name "phytases" but we did not perform tests to evaluate the three-dimensional structure of the enzyme. However, many authors refers a phytate degrading enzymes from E. coli as phytases (e.g. Menezes-Blackburn et al. 2011, doi:10.1016/j.biortech.2011.07.054; Derjsant-Li and Kwakernaak 2019, doi: 10.1016/j.anifeedsci.2019.05.018). Then, we believe that it is correct to use the term phytase to describe enzymes that degrade phytates from *E. coli* and *A. niger*.

Taking into account this explanation, we enlarge the introduction to clarify this issue. Now read as:

"Phytases are enzymes released by bacteria, fungi, plants and animals (Jorquera et al., 2008) and are able to catalyze the release of P from phytates. Phytases have the ability to release the 6 Pi molecules that are contained in phytate (Misset 2003)."

An alternative title may be: "Adsorption to soils and biochemical characterization of phytate-degrading enzymes (phytases)" (following Konietzny and Greiner 2002). We are ready to move for this title if the editor and reviewer consider this as a better option. Anyway, and we propose to maintain the original title "Adsorption to soils and biochemical characterization of purified phytases", since "a priori" we did not know the real capacity of the purified enzymes to release P.

...

2. The authors refer to the work of Misset 2003 as a reference of E coli phytases and their relevance in the industry. There are a couple of issues here. First I'm not really sure of the relevance of all strains of E coli phytases for the industry. If any which ones?.

R: Commercial phytases available in Argentina are purified from *A. niger* and E.coli, so we considered necessary to cite previous reports about E.coli as a source of phytase. Our E coli enzymes came from different commercialized products, mainly used for animal feed application. The strains are called with the same name as the product (TS Smizyme phytase, by Quimtia EDF, and Ronozyme, by DSM Nutritional Products Argentina S.A.). We used the same approach than Menezes-Blackburn et al. 2015 doi: 10.1021/acs.jafc.5b01996, who mention three *E. coli* strains that are isolated and commercialized. See below the table extracted from this paper

...

3. Many strains of *E. coli* possess an active phosphatase A gene witch can provide a certain level of phytate degradation but a real level of commercial degradation, I'm not sure about it

R: We agree. For this reason, we decided to perform these experiments to verify the actual activity of commercial enzymes purified from *E. coli* and *A. niger*.

...

4. Only until the lines 60 to 70, the really important point of the work was revealed. The main point in my perspective is the usage of phytases as biological fertilizers to re-lease inorganic P from organic P sources. But so far the whole history sounded more focused on something else. The major problem of the paper starts with the first hypothesis: Phytases have the ability to release P from different organic P sources, with a preference for phytic acid. In that way is redacted that is not a hypothesis that contributes at all with new knowledge in the field. It is already known that some phytases are highly specific and others are not but have preferences for phytate. Similar to the other two. Many references for that just two examples: doi:10.1128/AEM.01384-15 doi:10.1128/mBio.01966-18

R: OK, we understand that our hypothesis in the original version can lead to misunderstandings. We are not talking about phytases in general, but specifically about commercial enzymes of our work.

First hypothesis now read as ... "four commercially available phytase products tested in this work have the ability to release P from different organic P sources, with preference for phytic acid".

...

5. Is the norm of the journal to include only some of the line numbers? That makes it more difficult for review.

R: Yes, we use the journal template for submitting the paper.

...

6. The abstract is very misleading because implies that the authors isolated the phytases from the fungi by themselves. That is not the case.

Line 18-19: The proportion of phytases found in the solid phase of the soil 60 minutes after addition was lower than that found in the liquid phase (23-34% vs.66-77%). This result is not well connected in the abstract, is coming out of nowhere.

R: OK, we rearranged the paragraph.

Abstract now read as …" Four purified phytases isolated from *Aspergillus niger* and *Escherichia coli* were characterized biochemically and in terms of their adsorption to soils belonging to the Mollisol order. Three different organic P substrates were used to measure enzyme activity in a wide range of pH (2.3 to 9) and temperatures (-10° to 70°C): p-nitrophenyl-phosphate (pNP), glyceraldehyde-3-phosphate (G3Phospahte) and phytic acid. Phytases have low affinity for solid phase (23.34% of adsorption after one hour of incubation. Phytases from *A. niger* showed a higher capacity to release P (13% on average), than phytases from *E. coli*. All phytases were active throughout the pH and temperature ranges for optimum crop production. The amount of P that *A. niger* phytases release at pH that is commonly found in agricultural soils (5.5-7) is as follows: pNP > phytic acid > G3Phosphate, whereas in *E. coli* phytases the order was pNP / phytic acid > G3phosphate. Obtained results are 20 promising in terms of the use of phytases as a complement to P fertilization in agricultural settings and encourages further studies under field conditions.."

...

7. Lines 38-39: There are different forms of inositol-phosphates and the most abundant from phytate (refers only to the salt form). But what exactly is the meaning of phytates in these lines and in the subsequent text in general?.

R: In our work, we want to test the ability of commercial products to release P from different P organic sources, so in this paragraph we introduce the different forms of organic P found in the soil and in what proportion they are found. Please take into account our reply to comment 1, in which we explain that the definition of phytases is enlarged in the new version.

•••

8. Line 48. E coli and the rest of the text please italicize where required.

R: OK: Done

...

9. The hypotheses are not real hypotheses in the way their current state. It is already known that phytases can use different substrates. The number two was proved by a paper that the authors cite https://doi.org/10.1002/jpln.201600421. Finally, the hypothesis number 3 is way too basic for being a good work hypothesis.

R: We understand the point raised by the reviewer. The original hypotheses may appear as basic for ultrapurified enzymes or recombinant proteins produced for academic or related activities. Previous studies about phytases mainly come from ultrapurified enzymes like the ones provided by lab products retailers such as Sigma and are used in academic labs. It is clear that this high quality but extremely expensive products cannot be used in real agricultural settings by farmers. In this report we tried to test at which extent commercial phytases have comparable performance than the ultrapurified materials. We think that it is not correct to extrapolate results from both type of products. In such sense, in the new version we will modify the text highlighting the commercial nature of our evaluated phytases. Anyway, we will modify the hypotheses by clarifying that we refer to these commercial enzymes.

Hypothesis now read as... " i) the four commercially available phytase products tested in this work have the ability to release P from different organic P sources, with preference for phytic acid, but differ in the pH and temperature levels to reach their optimum activity ii) the retention of commercial phytases in the soil solid phase is associated to the soil clay content.

...

10. The biochemical characterization needs to include the catalytic efficiency of the reactions.

R: Done. Information was added to fig. 4

•••

11. It has been demonstrated recently by the works of Tan et al (doi:10.1007/s00253-015-7097-9) and others in 2019 using metagenomes that phytases are also present in metagenomes of soils. In fact, their presence is underestimated. Where is the experiment which proves that the used soils have low phytase activity? The control reactions of the initial experiments are missing.

R: Very good observation. In all experiments, we use blank reactions to ensure that the results presented in this work are those observed by the interaction of the enzymes with the soil. We measure soil phytase activity without the addition of the enzyme and observe soil phytase activity less than 1nkat g soil -1.

This topic was not clear enough in the previous version of the ms. Materials and methods now read as... "Phytase activity of the soil suspension was calculated as the difference between the soil suspension with enzyme minus the soil suspension without enzyme."

...

12. Line 134 I don't think is a good idea to use a demo or student versions of any software for statistical analysis in a publication.

R: Table Curve Demo version gives a limited period of software use (30 days), but it has the same mathematical functions as the full version. Anyway, we agree to remove the "demo version" if required. Statistical analysis were performed with the Statistix student version which is the software that we used in our lab since long time ago and was tested several times. For this particular paper and following this comment, we performed again all our analysis with INFOSTAT software (https://www.infostat.com.ar/) and we did not obtain different results. This software is cited in the new version.

...

13. Were the buffers set at the optimal temperature?

To evaluate the performance of the enzymes along a pH range (2.3-9.0), 200  $\mu$ l of each enzyme solution was diluted with 400  $\mu$ l of 50 mM glycine-HCl buffer (pH 2.3-4.4),

R: The buffers were prepared at normal room temperature (20-24 °C) and the incubation experiments were performed at 25 °C (for evaluating optimum pH and kinetic parameters) and along a temperature range (-10-70°C) for evaluating optimum temperature. In this experiment pH was set as 5.5. For these experiments we followed the approach proposed by George et al. (2005) and Hayes et al. (1999). The original text was somewhat unclear at this point and we reworded the ms accordingly.

...

14. The authors refer at the begging of the manuscript to the type of enzymes as 3-phytases. But they do not mention what type of enzymes are from the structural point of view. Are they acid phytases? Maybe that's is why need pH relatively low to act. But nothing of this is mentioned in the text. Is the optimal pH was determined before that the temperature is obvious that they did not set the buffer for the pH test at the right temperature. Therefore the pH characterization is not trustworthy.

**R:**

Commercial phytases used as a complement to poultry nutrition must be active at the stomach pH of the animals (about pH 3). Phytases that are active at that pH value are acid phytases, so in the introduction we mention the 3-phytases and 6- phytases that are by definition acid phytases (lines 41-50 of new paper version, *"A. niger* phytases are mainly extrinsic (Azeem et al., 2015), and are classified as 3-phytases, because they primarily dephosphorylate the phosphate group located at 3-position. *E. coli* phytases are mainly membrane-associated proteins and were classified as 6-phytase (Azeem et al., 2015). The classification as 3- or 6-phytases is related to which phosphate group is attacked first and would be determined by conformational differences in the  $\beta$ -domain of each phytase (Konietzny and Greiner, 2002)"). This is not a paper on poultry nutrition so we made no mention of this topic in the introduction section. It is possible that when we did the pH assay, the buffers were not set at the optimal temperature of the enzyme, but we decided to perform the assays according to Hayes et al. (1999) in order to make our results comparable with the literature. Anyway, as it can be seen in the results section, optimal temperature of function of the enzymes is close to the values worked in the experiments (20-29°C).

...

15. It seems that the authors did not review any literature about phytases in 2019.

R: We checked all papers on phytases published in the top journals before the submission date of our SOIL ms. To the best of our knowledge we cited the most relevant paper but probably we missed some.

.....
1. Comments for editor. The research work carried out under the theme "Adsorption to soils and biochemical characterization of purified phytases" is of scientific significance and has practical application for release of Pi from native or exogenously added organic P. Though, the study conducted is well organized but certain points need due attention. The Accession no of microbial strains used in the study is missing.

R: Thanks for your comment; in our experiment we used four commercial phytases. Two came from two different batches of *A. niger* commercially sold under the name "Habio phytases", which were obtained from Sichuan Habio Bioengineering Co.Ltd (Sichuan, China). The other two, came from two strains of *E. coli*. One is sold under the name "TS Smizyme phytases", obtained from Quimtia EDF (Buenos Aires Argentina) and the other is sold under the name "Ronozyme", obtained from DSM Nutritional Products Argentina S.A. Unfortunately, and as usual for commercial strains, no accession number was provided by the supplier. Anyway, we rearranged the paragraph to provide all available information.

Materials and methods now read as (New ms lines 68-73):

"Two phytases isolated from *A. niger* and two from *E. coli* were used in our experiments. In the first case, here named *A. niger* 1 and 2, phytases came from two different batches of the fungus which are commercially sold under the name "Habio phytases" by Sichuan Habio Bioengineering Co.Ltd (Sichuan, China), In the E.coli group, the first selected enzyme (here called *E. coli* 1) is sold under the name "TS Smizyme phytase", by Quimtia EDF (Buenos Aires Argentina), and the second (here called *E. coli* 2) is sold under the name "Ronozyme", by DSM Nutritional Products Argentina S.A."

....

2. The cost incurred on purchase of purified phytases and their availability needs mention.

R: Enzymes were provided for free by the different companies producing and / or importing enzymes in the country. This information is included in the new version.

Materials and methods now read as (New ms line 73):

"These enzymes are in powder format at a concentration of 5000 U g-1 and was provided free of charge by the companies that produce or import them."

3. A comparative study with crude phytase obtained from wild strains of *A. niger* and *E. coli* could have also been conducted alongside.

R: This would be a good option for the next phase of this investigation. Although it was was not the main objective of the present study. Anyway, this comment is highly appreciated and will be taken into account in our next phase.

••••

4. Technical Comments for authors

Abstract needs some modification indicating the % increase in P release with A. niger over E. coli.

R: Done.

Abstract now read as (New ms line 12):

"Phytases from A. niger showed a higher capacity to release P, than phytases from E. coli (+13% on average)."

....

5. L-13-14 Please shift substrates pNP, G3phosphate and phytic acid after substrates

R: Done.

Abstract now read as (New ms line 11):

"..... p-nitrophenyl-phosphate (pNP), glyceraldehyde-3-phosphate (G3Phospahte) and phytic acid."

...

6. L-16 Please write that the order of P release from different substrates by *A. niger* and *E. coli* followed this trend (mention the trend).

R. OK. lif we understood the meaning of the comment, the order is mentioned a couple of lines above.

Abstract now read as (New ms lines 13-15):

"All phytases were active throughout the pH and temperature ranges for optimum crop production. The amount of P that *A. niger* phytases release at pH that is commonly found in agricultural soils (5.5-7) is as follows: pNP > phytic acid > G3Phosphate, whereas in *E. coli* phytases the order was pNP / phytic acid > G3phosphate".

•••

7. Introduction

L-24 Delete appropriate

R: OK.

Introduction now read as (New ms line 20):

"Most strategies for enhancing P nutrition of agricultural crops aim to maintain soils at the convenient P critical level so that yields..."

....

8. There are approximately 38 references in introduction. The no. can be reduced.

R: OK. In the new version, the number of references was reduced to 22.

....

9. L-45-48. The first phytase was discovered—– delete this paragraph.

R: OK, deleted.

••••

10. L-74 instead of level write pH and temperature optima.

R: OK.

Introduction now read as (New ms line 65):

... " the two evaluated phytases differ in their optimum pH and temperature to reach their maximum activity ..."

....

11. Material and methods

L-77 A. niger in italics

L-79 powder form and not format

L-81 Superscript g-1

R. OK, all these editing issues were arranged as suggested.

....

12. L-87 If one g soil was mixed with 20 ml phytase solution how can you take a sub sample of 500 ml. Please check the unit

R: We apologize because it was our mistake. The subsample volume is 500 microliters (not ml). Arranged in the new version.

Materials and methods now read as (New ms line 84):

"...sub-samples of soil slurry (500  $\mu$ l) were taken for phytase activity measurements..."

....

13. L-92 150 ml or 150 microliter

R: Correct, same as above, it is 150 microliter. Arranged in the new version.

Materials and methods now read as (New ms line 86):

"An aliquot (150  $\mu l)\,$  of the soil slurry was used to measure the enzyme activity..."

••••

14. L-105 total protein (Lowry et al.)

R. OK.

Materials and methods now read as (New ms line 100):

"Biochemical characterization of the phytases included: total protein (Lowry et al., 1951)"

....

15. Phytase activity was measured with 3 substrates

R: OK, arranged as suggested

Materials and methods now read as (New ms line 102):

"Phytase activity was measured with 3 substrates containing..."

....

16. L-119 Blanks for measuring enzyme activity included (i) (ii) (iii)

R: OK.

Materials and methods now read as (New ms lines 112-113):

"The activities were tested against three blanks: (i) reaction buffer without enzyme or substrate; (ii) reaction buffer with enzyme without substrate; and (iii) reaction buffer without enzyme with substrate."

••••

17. L123-126 Please rewrite this portion

R: OK.

Materials and methods now read as (New ms line 114):

"For the pNP substrate, the enzymatic activity was measured at 412 nm which is the absorbance value of pnitrophenol (Hayes et al., 1999). The concentration of 3 substrates was determined as the concentration of the whole sample minus the concentration of the reaction blank."

Please check that the sentence: "Phytase activity with phytic acid and glyceraldehyde-3-phosphate as substrates was measured as P release measured by the 125 Murphy-Riley method (Murphy and Riley, 1962)" was eliminated because this procedure is provided in the previous sentence.

....

18. Please mention the amount of TCA added to stop the reaction

R: OK.

Materials and methods now read as (New ms line 91):

"Reactions were stopped with an equal volume of 10% TCA (300  $\mu l$  in soil slurry experiments and 700  $\mu l$  in soil solution experiments)."

....

**19. L-192 Modify the sentence**

R: OK.

Results now read as (New ms line 180):

"All four enzymes were effective in releasing P from phytic acid throughout the entire pH range analyzed"

....

20. L-200 pH 7.8 was detrimental for release of Pi from pNP by A. niger

R: OK, arranged as suggested.

Results now read as (New ms line 186):

"pH 7.8 was detrimental for release of Pi from pNP by *A. niger*, probably because the hydrolysis of the substrate

....

21. L-216 Change offered to tested substrates

R: OK.

Results now read as (New ms line 202):

"A. niger showed maximum activity at 24 °C (Fig. 3a), releasing 33% of the original P contained in the substrate."

••••

22. Results and discussion

Discussion part is totally missing and needs to be written properly

R: OK, results and discussion sections were now written separately so as not to create confusion for the reader.

••••

23. No explanation for findings is given Conclusion needs to rewritten.

R: OK, conclusion section was reworded accordingly. After reorganizing the ms and split the Results and Discussion section into to separate parts, the conclusion was incuded in the last paragraph of the discussion. In this paragraph the test of hypothesis is specifically considered

Conclusion now read as (New ms lines 281-295):

"Obtained results partially support our first hypothesis since the selected phytases showed a great ability to release P from different organic P sources, but A. niger 1, 2 and E. coli 1 release more P from pNP than phytic acid while E. coli 2 has no preference for any particular substrate. Regarding to activity of phytases at different pH and temperature levels, phytases exhibited some differences in their pH and temperature levels to reach their optimum activity In contrast, our results did not support the second proposed hypothesis, since the retention of phytases by the soil solid phase did not have a clear association with the analyzed soil properties. In this regard, it must be taken into account that the seven selected soils belonged to the Mollisol order. After being added to the soil, tested phytases showed an adsorption to soil solid phase ranging from 20 to 40%. Those phytases that remain in the solution could release Pi from the organic P of the soil, whereas phytases that remain adsorbed to the soil solid phase could be released later. All studied phytases remained active at the optimum soil pH range of the most productive agricultural soils (5-7). In the same line, optimal temperatures for phytase activity were also within the temperature range more suitable for most agricultural crops (20-30°C). Our results suggest that purified phytases may constitute a feasible tool to be used as a complement to P fertilization. In such sense, further experiments should be performed to evaluate the enzyme performance under field conditions to evaluate the ability of phytases to release from organic soil P sources, their interaction with soil microorganisms and to test if crops can capitalize the eventual provision of inorganic P released."

....

23. Tables and Fig titles need to be precise

R: OK, all titles were rewritten

Tables and figures now read as:

"Table 1. Characteristics of seven representative soils of the Argentina's Pampa Region used for testing phytases adsorption. Samples were taken at 0- 20 cm, air dried and sieved at 1 mm prior to the analysis.

Table 2. Coefficients of the adjusted Lorentizian-peak functions for phytase activity (see graphs in Fig. 2) at different pH levels with phytic acid, p-nitrophenyl-phosphate and glyceraldehyde-3-phosphate as substrates. The equations were adjusted from the observed results of the release of P from each substrate used. Four purified phytases (two isolated from *A. niger* and two from *E. coli*) were evaluated. In those cases where significant differences between enzymes (analyzed by F tests) were not found, a unique curve were fitted.

Different letters correspond to significant differences between treatments (P < 0.05, LSD procedure) Coefficient a is the maximum percentage of P released; b is the pH value where the enzyme has maximum activity (a P release peak); c estimates the standard deviation of the distribution and x is the pH value.

Table 3. Coefficients of the adjusted Lorentizian-peak functions for phytase activity (see graphs in Fig. 2) at different temperature levels with phytic acid, p-nitrophenyl-phosphate and glyceraldehyde-3-phosphate as substrates. The equations were adjusted from the observed results of the release of P from each substrate used. Four purified phytases (two isolated from *A. niger* and two from *E. coli*) were evaluated. In those cases where significant differences between enzymes (analyzed by F tests) were not found, a unique curve were fitted. Different letters correspond to significant differences between treatments (P <0.05, LSD procedure) Coefficient a is the maximum percentage of P released; b is the temperature value where the enzyme has maximum activity (a P release peak); c estimates the standard deviation of the distribution and x is the temperature value.

FIGURE 1. Phytase activity distributed in the liquid and solid phases for the phytase soil adsorption experiment. Four purified phytases (two isolated from *A. niger* and two from *E. coli*) were evaluated. Experiments were performed with the seven soils described in Table 1. For *A. niger* 1 and 2 and *E. coli* 1 phytases, a unique curve decay (Eq. 2), and exponential increase (Eq. 3) involving the seven soils was fitted because no significant differences (after F tests) were found between them. For *E. coli* 2, no function could be adjusted because a 37% binding to the soil solid phase was observed at 5 minutes and remained stable throughout the incubation period. Each point represents the average of three observations. Bars represent standard error of the mean.

FIGURE 2. Phytase activity measured at different pH levels with phytic acid, pNP and G3Phosphate as substrates. Four purified phytases (two isolated from *A. niger* and two from *E. coli*) were evaluated. In those cases where significant differences between enzymes (analyzed by F tests) were not found, a unique curve was fitted. Each point represents the average of three observations. Bars represent standard error of the mean. Coefficients of each adjusted model are shown in Table 2.

FIGURE 3. Phytase activity measured at different temperature levels with phytic acid, pNP and G3Pphosphate as substrates. Four purified phytases (two isolated from *A. niger* and two from *E. coli*) were evaluated. In those cases where significant differences between enzymes (analyzed by F tests) were not found, a unique curve was fitted. Each point represents the average of three observations. Bars represent standard error of the mean. Coefficients of each adjusted model are shown in Table 3.

FIGURE 4. Kinetic parameters for phytic acid, pNP and G3Pphosphate as substrates of purified phytases (two isolated from *A. niger* and two from *E. coli*). The activity was determined at different P concentrations (0 to 100 mM) contained in each substrate. Each point represents the average of three observations. Bars represent standard error of the mean. Data were fitted to a Michaelis-Menten curve and the estimated Vmax and Km values obtained by the Lineaweaver-Burk method are shown."

- 24. Table 1 Mg+2 and Ca+2 and not Ca+1
- 25. Provide space between C total, P total and P inorganic
- 26. Fig.1. spelling for enzymatic
- R: OK, all these editing issues were arranged as suggested.

**Adsorption to soils and biochemical characterization of purified phytases**

María Marta Caffaro1,2; Karina Beatriz Balestrasse1,3; Gerardo Rubio1,2

1INBA, CONICET UBA, Buenos Aires, C1417DSE, Argentina

2 Soil Fertility and Fertilizers, School of Agriculture University of Buenos Aires, Buenos Aires, C1417DSE, Argentina 3 Biochemistry, School of Agriculture University of Buenos Aires, Buenos Aires, C1417DSE, Argentina

Correspondence to: Gerardo Rubio (rubio@agro.uba.ar)

Abstract. Abstract. Four purified phytases isolated from Aspergillus niger and Escherichia coli were characterized biochemically and in terms of their adsorption to soils belonging to the Mollisol order. Three different organic P substrates were used to measure enzyme activity in a wide range of pH (2.3 to 9) and temperatures (-10° to 70°C): phytic acid, pnitrophenyl-phosphate (pNP), and glyceraldehyde-3-phosphate (G3Phosphahte)-and phytic acid., Phytases haved low affinity for the solid phase,: 23-34% of the added amount was absorbed of adsorption bed after one hour of incubation. Phytases from A. niger showed a higher capacity to release P (13% on average 36 to 50% of P contained in the substrates, 44 to 62 µg P), than phytases from E. coli (24 to 15 36%, 20 to 44 µg P). All phytases were active throughout the pH and temperature ranges for optimum crop production. At pH values commonly found in agricultural soils (5.5-7) A. niger phytases released P following the ranking of substrates The amount of P that A. niger phytases release at pH that is commonly found in agricultural soils (5.5-7) is as follows : pNP > phytic acid > G3Phosphate, 
[revised manuscript text omitted]

 adsorption tests of phytases to soils. The samples were taken at a depth of 20 cm, air dried and screened at 1 mm prior to

 the analysis. Characteristics of the seven soils (0-20 cm) used in the phytase adsorption experiment.

| Soil                  |                    | Alberti            | Adelia
María     | Lincoln            | Oliveros           | San
Antonio
de Areco | Balcarce           | Balcarce           |
|-----------------------|--------------------|--------------------|---------------------|--------------------|--------------------|----------------------------|--------------------|--------------------|
| Soil type             |                    | Typic
Argiudoll | Entic
Haplustoll | Typic
Argiudoll | Typic
Argiudoll | Typic
Argiudoll         | Typic
Argiudoll | Typic
Argiudoll |
| pH                    |                    | 5.9                | 6.3                 | 6.0                | 5.7                | 6.1                        | 6.5                | 5.9                |
| Ca ±2+     | cmol c  | 3.6                | 3.0                 | 3.0                | 2.5                | 9.1                        | 6.5                | 5.2                |
| $Ca^{+2+} + Mg^{+2+}$ | kg -1   | 4.5                | 4.0                 | 4.0                | 3.2                | 6.1                        | 7.1                | 5.6                |
| C T_ total | g kg -1 | 26.0               | 11.5                | 14.2               | 14.0               | 20.2                       | 38.6               | 36.9               |
| Clay                  |                    | 16.3               | 16.7                | 8.8                | 28.8               | 30.0                       | 27.6               | 36.4               |
| Sand                  | %                  | 44.0               | 51.3                | 68.0               | 8.3                | 19.4                       | 34.6               | 23.8               |
| Silt                  |                    | 39.8               | 32.0                | 23.3               | 63.0               | 50.6                       | 36.5               | 48.6               |
| P Bray 1   |                    | 14.9               | 16.2                | 3.4                | 14.9               | 3.4                        | 24.6               | 35.6               |

|   | P Mehlich 3   |                                                    | 20.3 | 19.3 | 12.9 | 20.8 | 6.9  | 36.1 | 48.6 |
|---|--------------------------|----------------------------------------------------|------|------|------|------|------|------|------|
|   | P T total     | mg kg -1                                | 351  | 308  | 284  | 290  | 228  | 441  | 453  |
|   | P o org           |                                                    | 208  | 148  | 150  | 181  | 163  | 339  | 325  |
|   | P I inorg     |                                                    | 142  | 159  | 134  | 109  | 64   | 102  | 129  |
| _ | $Al^{\pm3\pm}$           | mmolc                            | 1.0  | 0.7  | 0.5  | 0.7  | 0.8  | 1.3  | 1.8  |
|   | Fe +3+ | kg
1 mmol e        | 1.3  | 1.1  | 1.3  | 1.1  | 1.4  | 1.9  | 2.3  |
| _ | Clay ssa-BET  | $\frac{\text{kg}^{-1}}{\text{m}^2 \text{ g}^{-1}}$ | 12.6 | 9.8  | 3.5  | 13.7 | 31.4 | 20.5 | 32.5 |

Table 2. Coefficients of the adjusted Lorentizian peak adjusted-functions for phytase activity at different pH levels (see graphs in Fig. 2). The function was adjusted from the observed results of the release of P from each of the substrates used at different pH levels. The substrates used waseremeasured at different pH levels with phytic acid, pNPpNP-nitrophenylPhytic phosphate and G3PG3Phosphateglyceraldehyde 3 phosphate as substrates. Four purified phytases (two isolated from *A. niger* and two from *E. coli*) were evaluated. In those cases where significant differences between enzymes (analyzed by F tests) were not found, a unique curve were fitted. Different letters correspond to significant differences between treatments (P <0.05, LSD procedure).

| Adjusted function: $y = \frac{a}{1 + (\frac{x-b}{c})^2}$ |        |                             |       |      |  |  |  |  |
|----------------------------------------------------------|--------|-----------------------------|-------|------|--|--|--|--|
| Enzyme                                                   |        | Coefficients R 2 |       |      |  |  |  |  |
| Substrate: Phytic acid                                   |        |                             |       |      |  |  |  |  |
|                                                          | a      | b                           | с     |      |  |  |  |  |
| A. niger $1 + A$ .                                       | 36.6a  | 5.9a                        | 2.7a  | 0.73 |  |  |  |  |
| niger 2                                                  |        |                             |       |      |  |  |  |  |
| E. coli 1                                                | 30.1b  | 5.5ab                       | 4.2a  | 0.55 |  |  |  |  |
| E. coli 2                                                | 24.2c  | 4.7b                        | 3.8a  | 0.66 |  |  |  |  |
| Substrate: p-Nitrophenyl phosphate                       |        |                             |       |      |  |  |  |  |
| A. niger $1 + A$ .                                       | 49.96a | 6.2a                        | 1.03a | 0.79 |  |  |  |  |
| niger 2                                                  |        |                             |       |      |  |  |  |  |
| E. coli 1                                                | 36.88b | 5.8a                        | 1.96a | 0.70 |  |  |  |  |
| E. coli 2                                                | 24.16c | 6.0a                        | 1.54a | 0.77 |  |  |  |  |
| Substrate: Glyceraldehyde-3-phosphate                    |        |                             |       |      |  |  |  |  |
| A. niger $1 + A$ .                                       | 44a    | 3.9b                        | 0.7b  | 0.94 |  |  |  |  |
| niger 2                                                  |        |                             |       |      |  |  |  |  |
| E. coli 1                                                | 36.6b  | 4.1b                        | 0.8b  | 0.89 |  |  |  |  |
| E. coli 2                                                | 24.2c  | 6.0a                        | 1.5a  | 0.77 |  |  |  |  |

Table 3. Coefficients of the adjusted Lorentizian peak functions for phytase activity at different temperature levels (see graphs in Fig. 3). The function was adjusted from the observed results of the release of P from each of the substrates used at different temperature levels. The substrates used wereas levels with phytic acid, pNP, G3PhosphatepNP-nNitrophenyl-phosphate and G3Pglyceraldehyde-3-phosphate as substrates. Four purified phytases (two isolated from *A. niger* and two from *E. coli*) were evaluated. In those cases where significant differences between enzymes (analyzed by F tests) were not found, a unique curve were fitted. Different letters correspond to significant differences between treatments (P < 0.05, LSD procedure)

Coefficients of the adjusted functions for phytase activity (see graphs in Fig. 3) measured at different temperature levels with phytic acid, p nitrophenyl phosphate and glyceraldehyde 3 phosphate as substrates. Four purified phytases (two isolated from *A. niger* and two from *E. coli*) were evaluated. In those cases where significant differences between enzymes (analyzed by F tests) were not found, a unique curve were fitted. Different letters correspond to significant differences between treatments (P <0.05, LSD procedure).

| Adjusted function: $y = \frac{a}{1 + (\frac{x-b}{c})^2}$ |        |     |                |      |  |  |  |  |
|----------------------------------------------------------|--------|-----|----------------|------|--|--|--|--|
| Enzyme                                                   |        |     | $\mathbb{R}^2$ |      |  |  |  |  |
| Substrate: Phytic acid                                   |        |     |                |      |  |  |  |  |
|                                                          | a      | b   | с              |      |  |  |  |  |
| A. niger 1 + A.
niger 2                               | 33.47a | 24a | 13.12b         | 0.94 |  |  |  |  |
| E. coli 1+
E. coli 2                           | 24.53b | 29a | 21.61a         | 0.86 |  |  |  |  |
| Substrate: p-Nitrophenyl phosphate                       |        |     |                |      |  |  |  |  |
| A. niger 1 + A.
niger 2                               | 17.74b | 29a | 20.78a         | 0.97 |  |  |  |  |
| E. coli 1                                                | 22.18a | 29a | 19.49a         | 0.96 |  |  |  |  |
| E. coli 2                                                | 13.22c | 29a | 19.5a          | 0.95 |  |  |  |  |
| Substrate: Glyceraldehyde-3-phosphate                    |        |     |                |      |  |  |  |  |
| A. niger 1 + A.
niger 2                               | 10.05a | 24b | 42.03b         | 0.80 |  |  |  |  |
| E. coli 1                                                | 6.62a  | 30b | 36.34b         | 0.84 |  |  |  |  |
| E. coli 2                                                | 12.61b | 20a | 53.4a          | 0.43 |  |  |  |  |

**Figures**

FIGURE 1. Phytase activity distributed in the liquid and solid phases for the phytase soil adsorption experiment. Four purified phytases (two isolated from *A. niger* and two from *E. coli*) were evaluated. Experiments were performed with the seven soils described in Table 1. For *A. niger* 1 and 2 and *E. coli* 1 phytases, a unique curve decay (Eq. (2)) and exponential increase (Eq. (3)) involving the seven soils was fitted because no significant differences (after F tests) were found between them. For *E. coli* 2, no function could be adjusted because a 37% binding to the soil solid phase was observed at 5 minutes and remained stable throughout the incubation period. Each point represent the average of three observations minus the controls described in Materials and Method section. Bars represent standard error of the mean.

---

## Author Response (AR2)

March 11, 2020

Ref: MS No.: soil-2019-50.  Title: Adsorption to soils and biochemical characterization of purified phytases.Maria Marta Caffaro et al.

Topical Editor SOIL Journal

Dear Dr Jeanette Whitaker

We would like to thank you for your valuable suggestions The detailed responses and changes made to the original manuscript are given below and included in the new manuscript. Changes are highlighted in yellow. If additional modifications are required, please let us know.

Regards,

Dr. Gerardo Rubio
School of Agriculture - University of Buenos Aires

……………………………………

Detailed response to the Editor and the reviewers:

1. Title, reviewer 2 indicated the title needed changing and you have suggested an alternative. It seems from your discussion that replacing purified phytases with either "phytate-degrading enzymes" or "commercial phytases" would clarify the scope of the paper.

R: OK

New title read as:

"Adsorption to soils and biochemical characterization of purified commercial phytases"

…..

2. The abstract needs 1 or 2 sentences at the beginning to explain the context of the study, why is this study necessary and what is the relevance, before you go into the methods and what you have done.

R: OK. New abstract starts as:

"Commercial phytases are widely used in poultry production, but little is known about their potential use as biofertilizer for agricultural crops as an alternative to reduce the use of synthetic fertilizers"

…..

3. You have two hypotheses in the introduction, but these are not referred to again until the conclusions section. I would like to see them referred to in the methods and results/discussion sections so it is clear how your experiments test these two hypotheses.

R: Good point, done.

New M&M line 80 read as:

"To test hypothesis ii), soil samples (0-20 cm) were taken from…"

New M&M line 102 read as:

"To test hypothesis i), we performed the biochemical characterization of four purified phytases. This characterization included:…"

New R &D line 160 read as:

" Obtained results did not support the proposed second hypothesis, since the retention of phytases by the soil solid phase did not have a clear association with the analyzed soil properties, including the soil clay content. Therefore, it was possible to fit a single model after pooling the data of the seven sites (Fig. 1)."

And line 173:

"However, it should be taken into account that the seven Mollisols used in this work did not have a wide range of textures."

New M&M line 204read as:

"The proposed hypothesis i) is.therefore only partially accepted since although all four purified phytases had the ability to use the three substrates, they released more P from p-nitrophenyl-phosphate  than from phytic acid.."